# Natural variation in autumn expression is the major adaptive determinant distinguishing Arabidopsis FLC haplotypes

Jo Hepworth[1], Rea L Antoniou-Kourounioti[2], Kristina Berggren[3], Catja Selga[4], Eleri H Tudor[5], Bryony Yates[1], Deborah Cox[1], Barley Rose Collier Harris[1], Judith A Irwin[5], Martin Howard[2], Torbjörn Säll[4], Svante Holm[3]*, Caroline Dean[1]*

[1]Cell and Developmental Biology, John Innes Centre, Norwich, United Kingdom; [2]Computational and Systems Biology, John Innes Centre, Norwich, United Kingdom; [3]Department of Natural Sciences, Mid Sweden University, Sundsvall, Sweden; [4]Department of Biology, Lund University, Lund, Sweden; [5]Crop Genetics, John Innes Centre, Norwich, United Kingdom

**Abstract** In *Arabidopsis thaliana,* winter is registered during vernalization through the temperature-dependent repression and epigenetic silencing of floral repressor *FLOWERING LOCUS C (FLC)*. Natural Arabidopsis accessions show considerable variation in vernalization. However, which aspect of the *FLC* repression mechanism is most important for adaptation to different environments is unclear. By analysing *FLC* dynamics in natural variants and mutants throughout winter in three field sites, we find that autumnal *FLC* expression, rather than epigenetic silencing, is the major variable conferred by the distinct Arabidopsis *FLC*haplotypes. This variation influences flowering responses of Arabidopsis accessions resulting in an interplay between promotion and delay of flowering in different climates to balance survival and, through a post-vernalization effect, reproductive output. These data reveal how expression variation through non-coding cis variation at *FLC* has enabled Arabidopsis accessions to adapt to different climatic conditions and year-on-year fluctuations.

*For correspondence:
Svante.Holm@miun.se (SH);
caroline.dean@jic.ac.uk (CD)

**Competing interests:** The authors declare that no competing interests exist.

## Introduction

Developmental transitions in plants are aligned with specific seasons to maximise reproductive success. A major seasonal cue used to time the transition to flowering is temperature (*Hancock et al., 2011*; *Andrés and Coupland, 2012*; *Manzano-Piedras et al., 2014*). Many plants overwinter in a vegetative form and flower in spring. This alignment of the floral transition with spring is achieved through a process called vernalization, in which flowering is accelerated by previous exposure to prolonged winter cold. In *Arabidopsis thaliana* and its relatives, FLOWERING LOCUS C (FLC), a MADS-box transcription factor, delays the transition to flowering in warm temperatures until its expression is downregulated by exposure to cold (*Michaels and Amasino, 1999*; *Sheldon et al., 1999*). On return to warm, spring-like conditions, *FLC* remains epigenetically silenced, permitting expression of *FT* and floral meristem identity genes that trigger the floral transition (*Helliwell et al., 2006*; *Searle et al., 2006*).

The vernalization mechanism has been genetically dissected using controlled laboratory conditions (*Song et al., 2013*; *Bloomer and Dean, 2017*; *Whittaker and Dean, 2017*). *FLC* expression is upregulated by the coiled-coil protein FRIGIDA (FRI), which establishes an active transcription state at *FLC* through interactions with chromatin modifiers including the H3K36 methyltransferase SET

DOMAIN GROUP8 (SDG8) (*Choi et al., 2011*; *Hyun et al., 2017*; *Li et al., 2018*). This upregulation is antagonised by the autonomous pathway, a group of interacting RNA-processing factors, (eg. FCA and FY) and chromatin modifying factors (eg. FLD and FVE). These promote a low expression state at *FLC* through interactions with the long non-coding antisense transcript group, *COOLAIR* (*Ausín et al., 2004*; *Liu et al., 2007*; *Wu et al., 2016*; *Wu et al., 2020*). A high *FLC* transcription state is associated with plants that overwinter before flowering. Cold transcriptionally downregulates *FLC,* and the locus is maintained in an epigenetically silenced state by a Polycomb Repressive Complex 2 (PRC2) mechanism (*Gendall et al., 2001*; *Bastow et al., 2004*; *Sung and Amasino, 2004*; *Greb et al., 2007*; *Helliwell et al., 2011*; *Hepworth et al., 2018*). This PRC2 mechanism requires the temperature-integrating accessory protein, VERNALIZATION INSENSITIVE3 (VIN3) (*Sung and Amasino, 2004*; *Bond et al., 2009a*; *Antoniou-Kourounioti et al., 2018*). Cold-induction of *VIN3* expression leads to formation of a PHD-PRC2 complex that also contains the PHD protein VERNALI-ZATION5 (VRN5) and the specific Su(z)12 component VRN2 (*Sung et al., 2006a*; *Greb et al., 2007*; *De Lucia et al., 2008*). This complex is nucleated at *FLC* by the B3-binding protein VP1/ABI3-LIKE 1 (VAL1) (*Qüesta et al., 2016*; *Yuan et al., 2016*). The PHD-PRC2 methylates histone H3 lysine 27 residues in a ∼ 3 nucleosome domain at *FLC* to epigenetically 'switch' each allele off in a stochastic manner that over the population of cells produces a quantitative memory of the length of cold (*Angel et al., 2011*; *Yang et al., 2014*; *Angel et al., 2015*; *Yang et al., 2017*). When plants return to warm conditions, the PHD-PRC2 spreads over the locus to maintain long-term epigenetic memory, and this requires the action of LIKE HETEROCHROMATIN1 (LHP1) (*Mylne et al., 2006*; *Sung et al., 2006b*; *Berry et al., 2017*; *Yang et al., 2017*).

*FLC* expression and memory is influenced by natural non-coding cis polymorphism at the locus (*Lempe et al., 2005*; *Shindo et al., 2006*; *Coustham et al., 2012*; *Li et al., 2015*; *Qüesta et al., 2020*). Different Arabidopsis accessions need different lengths of cold to epigenetically silence *FLC,* and this requirement is determined by non-coding SNPs at *FLC* (*Shindo et al., 2006*; *Coustham et al., 2012*). Indeed, within the Arabideae tribe, cis regulation of *FLC* orthologues contributes to perennial-vs.-annual flowering differences (*Kiefer et al., 2017*). Within the worldwide *Arabidopsis thaliana* population, non-coding variation defines a small number of *FLC* haplotypes, which confer quantitatively different flowering time responses to different lengths of vernalization in constant cold in the laboratory (*Li et al., 2014*). These haplotypes appear to have been maintained in the *A. thaliana* population due to their contributions to life history diversity (*Sánchez-Bermejo et al., 2012*; *Li et al., 2014*; *Méndez-Vigo et al., 2016*), and they show different geographic distributions, suggesting that they are adapted to different climates (*Méndez-Vigo et al., 2011*; *Li et al., 2014*).

Recent work has shown the importance of analysing flowering in natural field conditions (*Figure 1A*), 'in natura' (*Kudoh, 2016*). Such studies have highlighted how the influence of genes of major effect, including *FLC*, changes depending on the field environment (*Wilczek et al., 2009*; *Fournier-Level et al., 2011*; *Duncan et al., 2015*; *Ågren et al., 2017*; *Taylor et al., 2019*). In particular, the investigation of RNA levels in natura has identified genetic mechanisms in the field that were previously unknown from laboratory observations (*Antoniou-Kourounioti et al., 2018*; *Song et al., 2018*; *Nagano et al., 2019*). For example, field work in Sweden and the UK with the vernalization reference genotype Col *FRI*[SF2] and mutants in *VIN3* (*vin3-4*) has shown that initial VIN3-independent *FLC* downregulation requires cold temperatures, whereas VIN3-mediated epigenetic silencing requires the absence of daily temperatures above 15°C (*Hepworth et al., 2018*). Autumn conditions greatly influence when epigenetic silencing initiates, sometimes leading to temporal separation of the two phases, which in laboratory constant-condition experiments and some natural environments occur concurrently (*Hepworth et al., 2018*).

Here, we exploit field studies in three climatically distinct locations, over two years, to define the roles of *FLC* cis polymorphism and known genetic regulators of *FLC* in VIN3-dependent and -independent phases of the vernalization response under natural conditions (*Figure 2A*). We hypothesised that accessions bearing different *FLC* haplotypes would show different flowering time responses to winter environments, and that this would be due, at least in part, to variation in the downregulation and silencing of *FLC* itself, arising from the cis polymorphism at *FLC*. Moreover, we expected variation in the *FLC* epigenetic silencing mechanism would play a particularly important role, as this mechanism has previously been shown to vary in natural accessions (*Coustham et al., 2012*; *Wollenberg and Amasino, 2012*; *Duncan et al., 2015*).

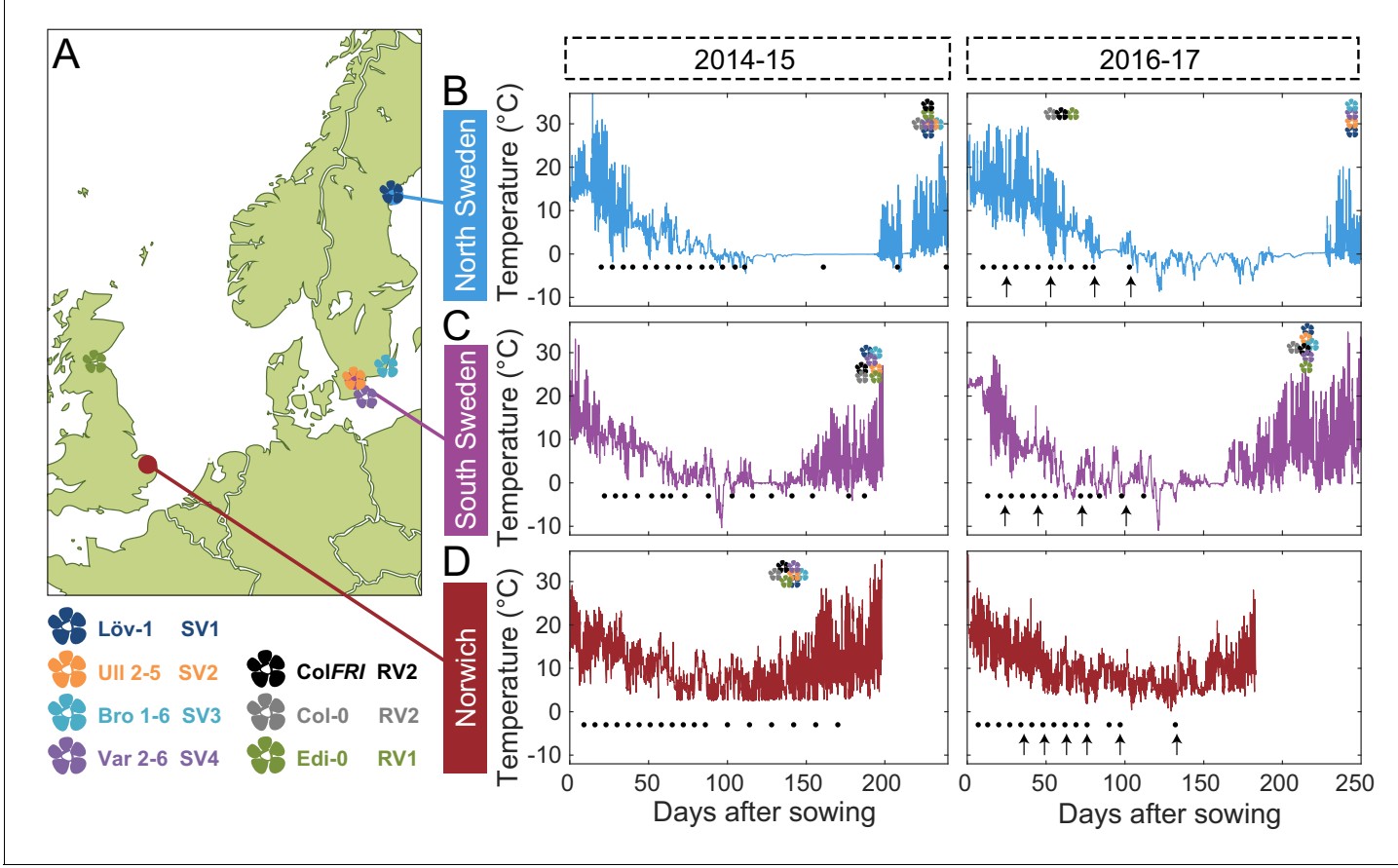

**Figure 1.** Field experimental setup. (**A**) Map showing locations of field sites (dots) and the origins of five of the accessions (flowers) used in this study. These accessions, with the addition of Col-0, represent the five major and one intermediate (Löv-1) *FLC* haplotypes identified by *Li et al., 2014*. The lab genotype Col *FRI* was also used in this study as a vernalization-requiring reference. (**B–D**) Temperature profiles experienced by plants at the three field sites, North Sweden – Ramsta (**B**), South Sweden – Ullstorp (**C**) and Norwich, UK (**D**) (*Source data 1*, as from *Hepworth et al., 2018* and *Antoniou-Kourounioti et al., 2018*). Flowers above temperature profile indicate the median time of bolting of each of the natural accessions and of Col *FRI* (legend at bottom left corner). Black dots below temperature profile indicate the timepoints when plant material was collected for expression analysis. Black arrows below temperature profiles indicate time of transfer to greenhouse with long-day, warm conditions to assess degree of vernalization based on bolting time.

The online version of this article includes the following figure supplement(s) for figure 1:

**Figure supplement 1.** Increased vernalization reduces time to bolting, variability in bolting time, and increases rosette branch production in different accessions.

**Figure supplement 2.** Increased vernalization reduces time to bolting and increased branch production with subtly different effects depending on *FLC* haplotype in the Col *FRI* background.

Our results demonstrate how in the field the major *FLC* haplotypes, differing only through non-coding variation, have different starting *FLC* levels and rates of response to autumn cold, but show more similar epigenetic silencing rates in winter. This results in effective repression of *FLC* by mid-winter and post-winter flowering in most years, across haplotypes and climates. Our experiments also reveal effects of the *FLC* haplotypes on prevention of precocious flowering in warm years, and on reproductive success after winter via effects on branching and silique number. By studying gene expression across years and climates, we have been able to dissect how non-coding cis variation at *FLC* modulates flowering time and fitness in response to different natural fluctuating environments.

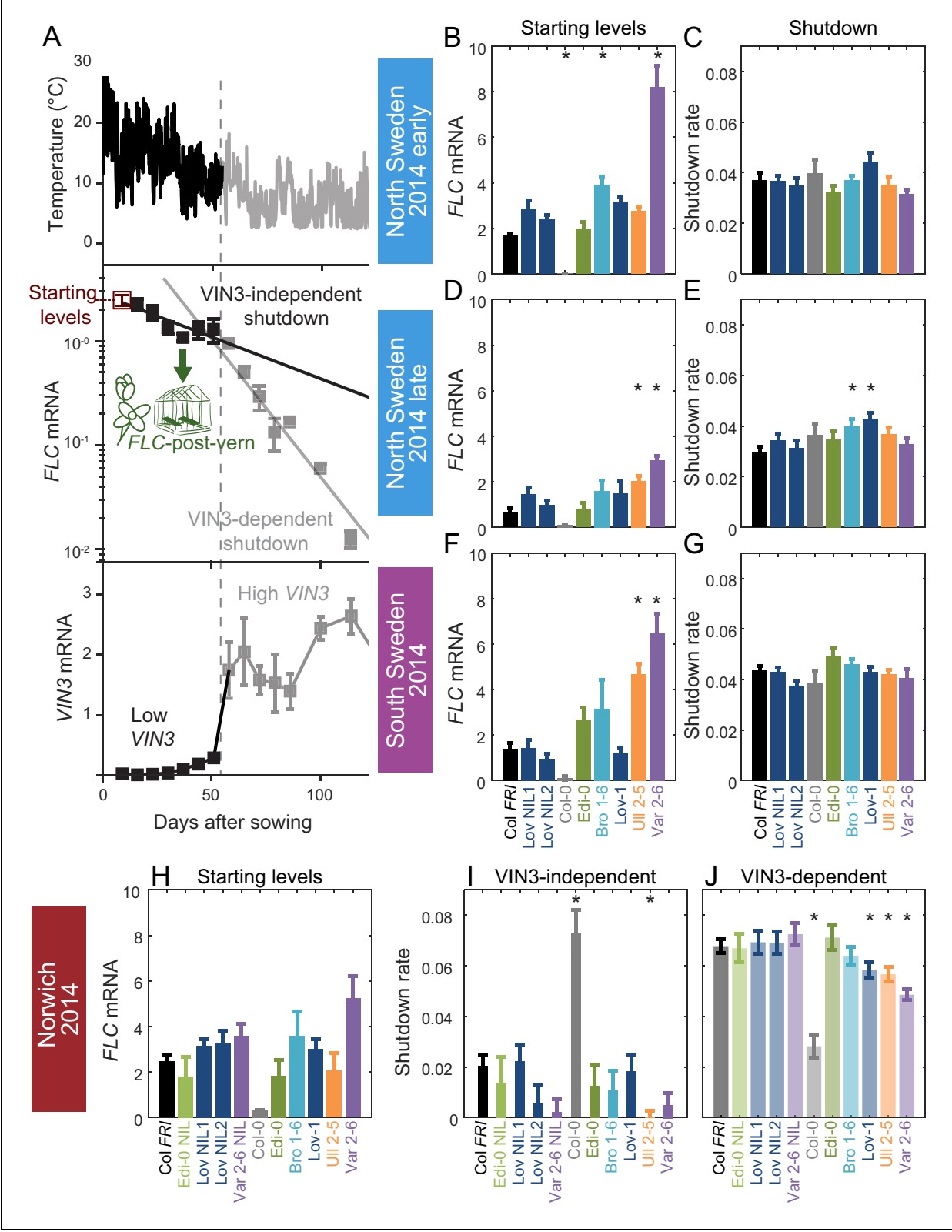

**Figure 2.** Downregulation in 2014–5 in Norwich, North Sweden (two plantings) and South Sweden for all NILs and accessions. (**A**) Experimental data for Col *FRI* in Norwich 2014–5, showing the temperature profile (top), *FLC* (middle) and *VIN3* (bottom) expression. Different shades indicate the separation of the VIN3-dependent (grey) and -independent (black) phases of *FLC* silencing (*Hepworth et al., 2018*) and equivalent times in *VIN3* and temperature profiles (as in *Figure 1D*). Expression data were normalised to the control sample for 2014–5 (see Materials and methods). N = 6 except where samples

*Figure 2 continued on next page*

*Figure 2 continued*

were lost to death or degradation (see Materials and methods and *Source data 2*). Error bars show standard error of the mean (s.e.m). The initial measurement in the field (Starting levels), the rate of downregulation before induction of *VIN3* expression (VIN3-independent, estimated from the slope of the fitted line) and the rate of downregulation after *VIN3* induction (VIN3-dependent) are the three features that were analysed and compared for each genotype and treatment in the next panels, based on the data of *Figure 2—figure supplement 1*. A new feature is also shown, the FLC-post-vern, that is measured based on the flowering time from plants transferred to glasshouses with inductive conditions and how that relates to the *FLC* levels at the time of the transfer. (**B–J**) *FLC* downregulation analysed as level at first time point (Starting levels), and rate of downregulation (Shutdown – combining early and later timepoints for *FLC* data – see Materials and methods) for North (**B–E**) and South Sweden (**F–G**), or rate of downregulation before (VIN3-independent, dark bars) and after (VIN3-dependent, translucent bars) *VIN3* induction for Norwich (**H–J**). Features of genotypes that are significantly different to the reference line Col *FRI* are indicated by * (for Starting levels, ANOVA with Dunnett's post-hoc test, for Shutdown rates, Satterthwaite's t-tests on REML Linear mixed model). p-values for all comparisons are given in *Supplementary file 1*. Rates of downregulation are given in units of 'a.u. per day', where the arbitrary units (a.u.) correspond to the normalised concentration of *FLC* mRNA, measured by qPCR. *VIN3* induction started at ~58 days in Norwich (*Figure 2—figure supplements 2–3*). Expression data were normalised to the control sample for 2014–5 (see Materials and methods). N = 6 except where samples were lost to death or degradation (see Materials and methods and *Source data 2*). Error bars show s.e.

The online version of this article includes the following figure supplement(s) for figure 2:

**Figure supplement 1.** *FLC* downregulation in accessions and NILs in Norwich, North Sweden and South Sweden 2014–5.
**Figure supplement 2.** *VIN3* upregulation in accessions in Norwich, North Sweden and South Sweden 2014–5.
**Figure supplement 3.** Expression of *VIN3* in NILs with the Col-0 *VIN3* allele in the field in 2014–2015.

## Results

### Field experiments

*Li et al., 2014* identified 20 *FLC* haplotypes across a worldwide panel of Arabidopsis accessions defined predominantly through non-coding SNPs. These haplotypes conferred different flowering time responses to vernalization at constant 5°C, with the accessions carrying them clustered broadly into 'Rapid Vernalizing' (RV) and 'Slow Vernalizing' (SV) types. To investigate their function in field conditions, we selected accessions to represent each of the five most populous haplotypes, representing more than 60% of tested accessions, as well as a further accession, Löv-1, for which there is evidence of local adaptation to the climate in the region of our North Sweden field site (*Duncan et al., 2015*; *Qüesta et al., 2020*). To compare these haplotypes in a common genetic background, we exploited extant and developed new near-isogenic lines (NILs) in which the *FLC* haplotype from each accession had been repeatedly backcrossed to Col *FRI*^SF2 ('Col FRI'; *Duncan et al., 2015*; *Li et al., 2015*). We confirmed that these selected accessions and NILs show variation in their rate of response to time in constant cold in the laboratory and also found that they varied in branch response to vernalization, an effect mediated by *FLC* (*Figure 1—figure supplements 1–2*, *Huang et al., 2013*; *Lazaro et al., 2018*).

These genotypes were tested across two years and three field sites, with the exception of three NIL lines, which were generated during the experiments (*Figures 1A* and *2B–J*). The three field sites in Norwich, UK, in Ullstorp, Sweden ('South Sweden') and in Ramsta, Sweden ('North Sweden') were chosen to represent climates with different winters. Norwich has a temperate oceanic climate and a mild winter, Ullstorp has a warm-summer continental climate with a winter in which temperatures are often below freezing, and Ramsta is subarctic, usually with snow cover during winter (hence the 0°C flatline in temperatures in *Figure 1B*; *Beck et al., 2018*). The Swedish sites are also close to the source sites of several of the tested accessions; Vår2-6 and Ull2-5 in Skåne, near or at Ullstorp; Löv-1 near Ramsta (*Figure 1A*). In the second year of experimentation, we also included the *vin3-1* mutant in the Col *FRI* background. The experiments ran from August/September 2014 until spring 2015 and again from August/September 2016 to the spring of 2017, with sowing times adjusted to each site (earlier in the colder climates). In the first year, two plantings were performed in North Sweden, two weeks apart. The temperatures that the plants experienced are shown in *Figure 1B–D*. We measured the levels of spliced and unspliced *FLC*, and mRNA levels of *VIN3*, to follow the progress of vernalization in the field. The transition to flowering (bolting) was assessed both in the field and by transfers to warm inductive conditions.

## Natural variation in different phases of *FLC* silencing in the field

Across all the genotypes we tested and all seven field experiments, as expected, *FLC* expression reduced over weeks in response to autumn and winter temperatures, and *VIN3* was upregulated. Previously, we had noted that in 2014–5 in Norwich, substantial *VIN3* upregulation did not occur until ~58 days after sowing, although temperature conditions were suitable for VIN3-independent *FLC* downregulation for most of this time (*Hepworth et al., 2018*). In the following field season, this pattern occurred again, with *VIN3* upregulation delayed until 48 days after sowing (*Antoniou-Kourounioti et al., 2018*). For the Col *FRI* reference, we had found that we could fit two separate exponential decay curves to *FLC*; the first for the initial, slow, VIN3-independent phase and the second for the faster, VIN3-dependent phase of the downregulation (*Figure 2A*; *Hepworth et al., 2018*). Thus three features of the *FLC* profile contribute to the level of *FLC* at any time: firstly the 'Starting level' of *FLC* before vernalization, secondly the rate of downregulation in the initial VIN3-independent phase, and thirdly the rate of VIN3-dependent downregulation (*Figure 2A*). This pattern was consistent across the accessions and NILs, allowing us to investigate the effect of natural variation on different aspects of *FLC* regulation. The time of upregulation of *VIN3*, and thus the time of switching from the VIN3-independent to the VIN3-dependent shutdown, also affects *FLC* levels, but this was very similar between genotypes at the same site.

In Norwich 2014–5, the RV *FLC* accession Edi-0 (haplotype group RV1) behaved very similarly to Col *FRI* (RV2). In comparison, the SV accessions (Löv-1, Ull2-5, Bro1-6 and Var2-6) show higher levels of *FLC* throughout the winter (*Figure 2*, *Figure 2—figure supplement 1*). Ull2-5 started with similar levels of *FLC* as Col *FRI*, but slower rates of downregulation in both the VIN3-independent and VIN3-dependent phases generated higher levels of *FLC* throughout autumn and winter. For both Var2-6 and its NIL, an apparently slower VIN3-independent rate of downregulation contributed to a higher *FLC* level, although the difference in rate is not significant. The VIN3-dependent phase was also slower in Var2-6 and Löv-1 (*Figure 2*).

In Sweden 2014–5, at both sites the VIN3-independent and -dependent phases occurred concurrently (*Figure 2*, *Figure 2—figure supplement 1*; *Hepworth et al., 2018*). There was little variation observed in the overall rate of *FLC* downregulation between genotypes, though the maximal *VIN3* induction was more variable (*Figure 2—figure supplement 2*), with the fluctuations depending strongly on complex responses to the temperature profile, as expected from analysis of Col *FRI*[SF2] (*Antoniou-Kourounioti et al., 2018*). Instead, most of the natural variation in *FLC* levels throughout winter in Sweden was generated by differences in the early expression level. In North Sweden again both RV accessions had similar Starting levels whereas the SV Swedish accessions were higher to different degrees. However, in South Sweden Löv-1 started with similar levels to Col *FRI*.

In the first year of field experiments we observed that a high proportion of the *FLC* shutdown occurred early in autumn, and furthermore, a lot of the variation in total *FLC* between lines was during that early phase. To maximise this period, and to extend the time that the plants spent in warmer autumn conditions, so mimicing potential climate change effects, we sowed the second field experiment earlier. For the 2016–7 experiment, we sowed plants in North Sweden and Norwich two weeks earlier and South Sweden three weeks earlier than for the 2014–5 season (*Figures 1A* and *3*, *Figure 3—figure supplements 1–2*). The differences in daylight length due to this earlier sowing must be considered in direct comparisons between years, but the timing of both sowing dates was within the natural germination range. In North Sweden, this early sowing was followed by an usually warm autumn: September temperatures at the nearest weather station, Lungö, averaged 11.04°C between 1990–2016, whereas they were 12.60°C in 2016, the fourth highest year during that period (*Swedish Meteorological and Hydrological Institute, 2020*, www.smhi.se/data/meteorologi/ladda-ner-meteorologiska-observationer). This produced a delay in *VIN3* induction similar to that seen in both years in Norwich (*Figure 1A*; *Antoniou-Kourounioti et al., 2018*). However, the first stage of shutdown was still more rapid in North Sweden than in Norwich, despite higher average temperatures in North Sweden (*Figure 1A*), so that both VIN3-independent and VIN3-dependent phases had similar rates, resulting in the appearance of a single decline. As in the previous experiment in Sweden, these rates of downregulation were generally similar between different genotypes, with higher *FLC* levels in SV accessions and the Var NIL due to higher starting levels (*Figure 3*, *Figure 3—figure supplement 1*).

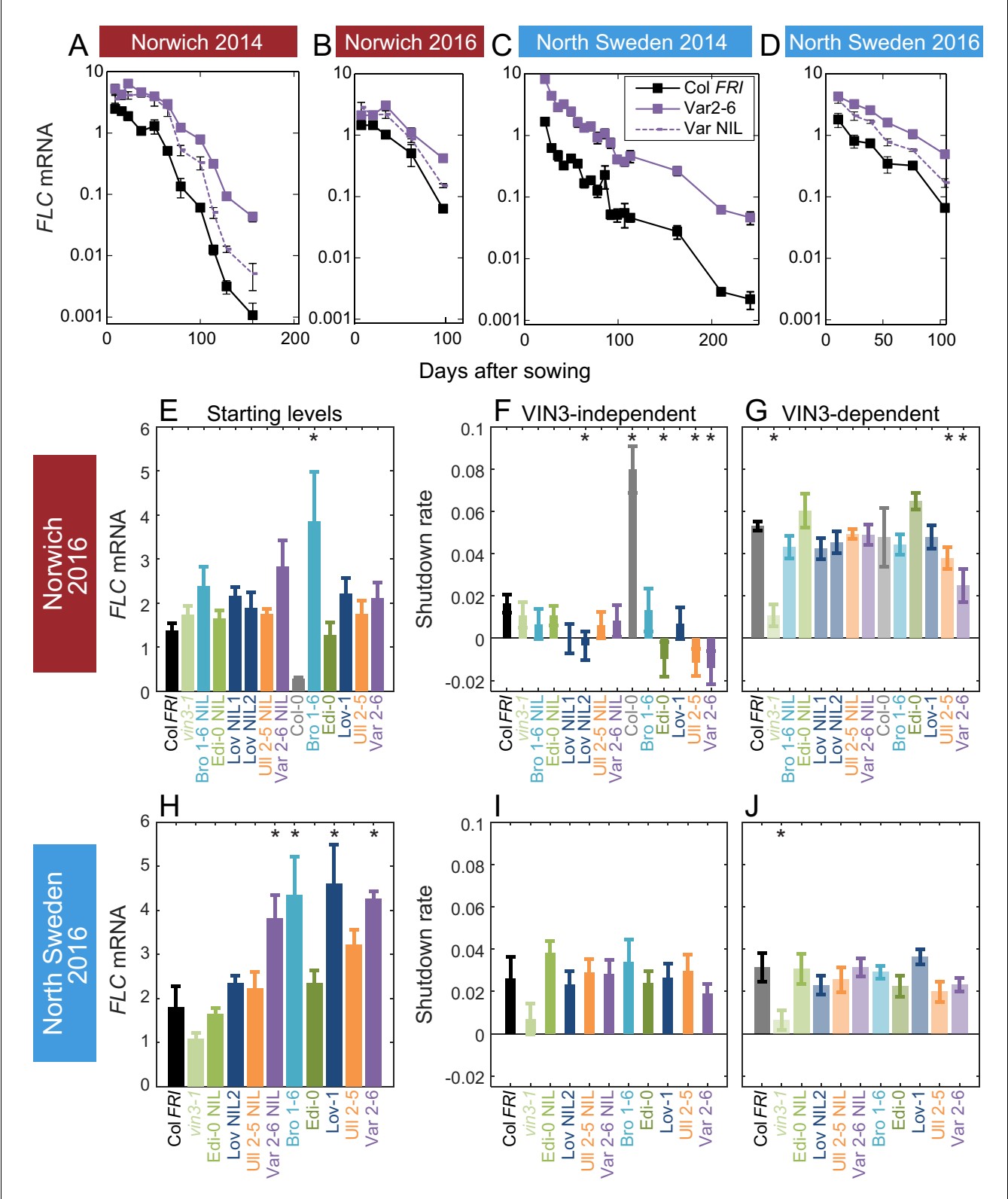

**Figure 3.** Downregulation in 2016 in Norwich and North Sweden for NILs and accessions show similar patterns of response to the first year. (A–D) *FLC* downregulation in Col *FRI*, Var2-6 and the Var NIL, as measured for Norwich and North Sweden in the winters of 2014–5 and 2016–7. (E–J) *FLC* downregulation as Starting level and VIN3-independent and dependent rates. Features of genotypes that are significantly different to the reference line Col *FRI* are indicated by * (for Starting levels, ANOVA with Dunnett's post-hoc test, for Shutdown rates, Satterthwaite's t-tests on REML Linear mixed

*Figure 3 continued on next page*

*Figure 3 continued*

model). p-values for all comparisons are given in *Supplementary file 1*. *VIN3* induction started at: Norwich 2016, ~48 days, North Sweden 2016,~46 days, see (*Figure 3—figure supplement 1*) Expression data were normalised to the corresponding control sample (2016–7, see Materials and methods). N = 6 except where samples were lost to death or degradation (see Materials and methods and *Source data 3*). Rates of downregulation are given in units of 'a.u. per day', where the arbitrary units (a.u.) correspond to the normalised concentration of *FLC* mRNA. Error bars of bar plots show s.e., of line graphs show s.e.m.

The online version of this article includes the following figure supplement(s) for figure 3:

**Figure supplement 1.** *FLC* downregulation and *VIN3* upregulation in accessions in Norwich and North Sweden in autumn/winter 2016.
**Figure supplement 2.** Downregulation of *FLC* and upregulation of *VIN3* in South Sweden in 2016.
**Figure supplement 3.** Low *VIN3* upregulation in Var2-6 is correlated with perturbation of the circadian clock.

In Norwich, the patterns seemed to be largely repeated, though our statistical power was lower in the 2016–7 experiment due to fewer timepoints. Bro1-6, Löv-1, Var 2–6 and their NILs appeared to have higher starting *FLC* levels, though of these only Bro1-6 was significant in our analysis. Ull2-5, Var2-6, Edi-0 and Löv NIL2 showed slower VIN3-independent downregulation and Ull2-5 and Var2-6 also again showed slower VIN3-dependent downregulation (*Figure 3*). Other than a slower VIN3-independent rate in Norwich in Edi-0, the RV haplotypes behaved similarly to each other in 2016–7.

The slower rate of the epigenetic silencing phase in Var2-6 was consistent across years in Norwich (*Figures 2* and *3*). Unlike the other accessions, this change in the VIN3-dependent phase in Var2-6 was consistently mirrored in lower levels of *VIN3* (*Figure 2—figure supplement 2*, *Figure 3—figure supplement 1*). The circadian clock is an important regulator of *VIN3* (*Antoniou-Kourounioti et al., 2018*; *Hepworth et al., 2018*), so we tested whether trans-variation within the clock contributes to the difference in *VIN3* upregulation in Var2-6. When sampled over 48 hr in the Norwich field experiment (*Figure 3—figure supplement 3*), *VIN3* expression in Var2-6 is much lower compared to our previous results from Col *FRI* (*Figure 3—figure supplement 3B,D*; *Antoniou-Kourounioti et al., 2018*), as is the expression of circadian clock component *CIRCADIAN CLOCK ASSOCIATED1*, the protein of which binds to the *VIN3* promoter (CCA1; *Figure 3—figure supplement 3*; *Nagel et al., 2015*). Therefore, variation in the circadian clock may underlie some of the difference in *FLC* regulation in Var2-6.

## Chromatin modifiers control *FLC* regulation in the field

To understand the contribution of other trans factors to temperature registration at the different phases of *FLC* shutdown in the field, we exploited our contained site in Norwich to investigate mutants in genes known to affect *FLC* levels before cold or in response to cold (*Figure 4*).

As previously reported (*Hepworth et al., 2018*), *vin3* mutants (*vin3-4, vin3-1*) did not show the increase in downregulation rate that marks the later epigenetic phase of silencing (*Figure 4C,F*). This effect was also seen for other mutants defective in epigenetic memory; *vrn1-4; vrn2-1* (defective in the Su(z)12 homologue component of PRC2); *vrn5-8* (defective in the VIN3-related protein VRN5); and *lhp1-3*, in accordance with this phase being required for epigenetic silencing. Loss of the H3K36 methyltransferase *sdg8* generated the same effect, correlated with its delayed upregulation of VIN3 (*Kim et al., 2010*; *Jean Finnegan et al., 2011*).

In the initial, VIN3-independent silencing phase, both Col *fri* (Col-0) and *sdg8 FRI* were hyper-responsive, consistent with a role for SDG8 with FRI in the establishment of the high *FLC* expression state (*Hyun et al., 2017*; *Li et al., 2018*). Mutants in the autonomous pathway, *fca-9, fld-4* and *fve-3*, which upregulate *FLC* expression in the absence of *FRI*, had no significant effect on VIN3-independent silencing, but behaved mostly like Col *FRI*, although *fve-3* reduced the rate of downregulation non-significantly in both years. The mutant with the most reduced VIN3-independent response was the B3-binding transcription factor VAL1, required for PRC2 action at *FLC* (*Figure 4E*; *Qüesta et al., 2016*; *Yuan et al., 2016*), followed closely by the *vrn* mutants, *vrn1-4* and *vrn5-8*, and a non-significant but consistent trend for *vrn2-1*. Conversely, the *lhp1-3* and *vin3* mutants do not show clear impairment in the early phase of downregulation. Overall, it seems that chromatin modifiers are required for both phases of downregulation in the field.

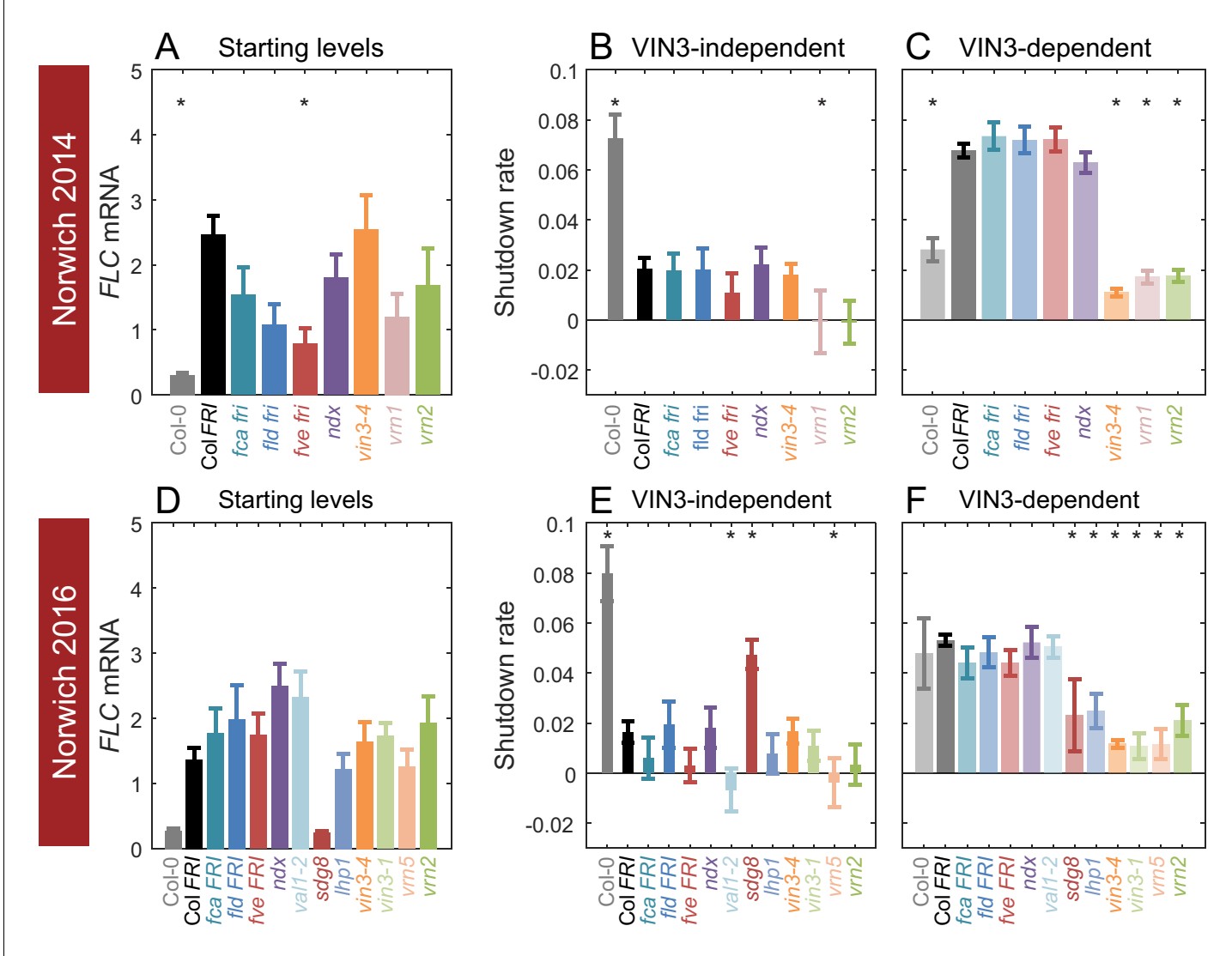

**Figure 4.** Starting levels and rates of downregulation of *FLC* in mutants and transgenics in field conditions in Norwich, UK. (A–F) *FLC* downregulation analysed as level at first time point (Starting levels, A, D), rate of downregulation before induction of *VIN3* expression (VIN3-independent, dark bars, B, E) and rate of downregulation after *VIN3* induction (VIN3-dependent, translucent bars, C, F). Features of genotypes that are significantly different to the reference line Col *FRI* are indicated by * (for Starting levels, ANOVA with Dunnett's post-hoc test, for Shutdown rates, Satterthwaite's t-tests on REML Linear mixed model). p-values for all comparisons are given in *Supplementary file 1*. Rates of downregulation are given in units of 'a.u. per day', where the arbitrary units (a.u.) correspond to the normalised concentration of *FLC* mRNA. *VIN3* induction started at: Norwich 2014, ~58 days, see (*Figure 4— figure supplement 1*); Norwich 2016, ~48 days, see (*Figure 4—figure supplement 2*). All mutants are in the Col *FRI* background unless otherwise stated. Expression data were normalised to the corresponding control sample (for 2014–5 or 2016–7, see Materials and methods). N = 6 except where samples were lost to death or degradation (see Materials and methods and *Source data 2* and *3*). Error bars show s.e.

The online version of this article includes the following figure supplement(s) for figure 4:

**Figure supplement 1.** Expression of *FLC* and *VIN3* in all mutants in the field in Norwich 2014–2015.

**Figure supplement 2.** Expression of *FLC* and *VIN3* in all mutants in the field in Norwich 2016–2017.

## Autumnal expression of *FLC* is the major variable in vernalization response

To identify which of the three features of *FLC* regulation is the major variable that defines the different haplotypes in different climates, we estimated the coefficient of variation for the rates of shutdown and starting levels for all the natural accessions, NILs, and *vin3-1* where available (from *Figure 2*, *3*, *Figure 2—figure supplement 1*, *Figure 3—figure supplement 1*, *Supplementary file*

*2*). In Sweden, the starting levels are significantly more variable than the shutdown rates (p-value=$2.8 \cdot 10^{-8}$ for the first year, *Figure 5A*), but in Norwich, the VIN3-independent shutdown rate is more variable than the starting levels and VIN3-dependent rate (p-value=$2.1 \cdot 10^{-5}$ for 2014, p-value=$5.0 \cdot 10^{-4}$ for 2016, *Figure 5B*). Combining Norwich and Sweden data, the early shutdown rate is again most variable (p-value=$1.9 \cdot 10^{-10}$ for 2014, p-value=$2.0 \cdot 10^{-5}$ for 2016, *Figure 5C*). On the other

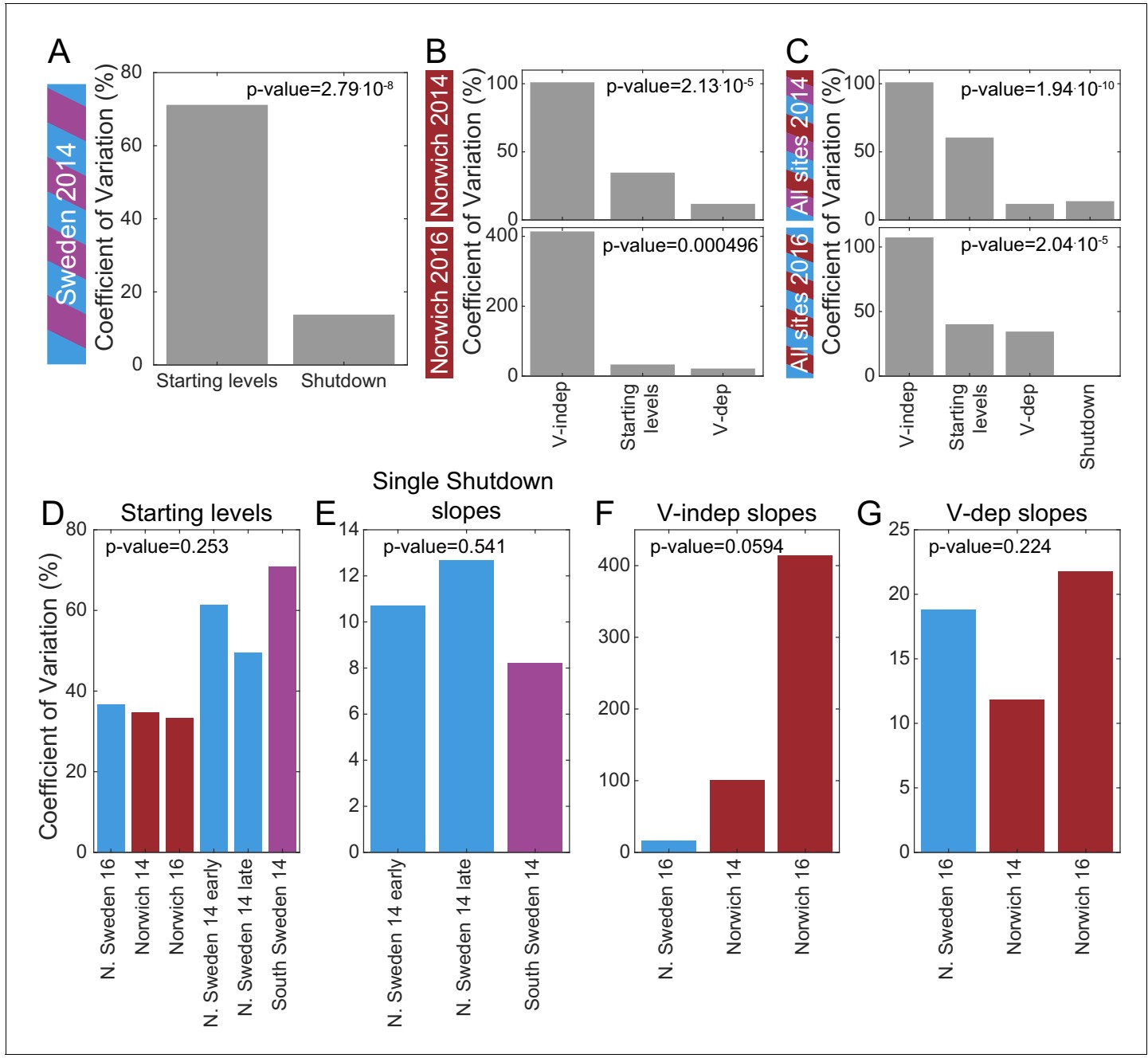

**Figure 5.** Mechanistic sources of natural variation in *FLC* levels across sites and years. (**A**) The coefficient of variation for the rates of shutdown and for the starting levels in all Sweden experiments in the first year. (**B**) Similarly in Norwich 2014 (top) and 2016 (bottom) but separately for the VIN3-independent (V-indep) and VIN3-dependent (V-dep) shutdown rates. (**C**) Comparison of the variability of the starting levels and the shutdown rates, separating V-dep and V-indep where appropriate, combining data from all sites in 2014 (top) and 2016 (bottom). 'Shutdown' refers to the combined V-dep/V-indep shutdown rate that was fitted in Sweden 2014, and so is not present in the 2016 results. (**D**) The coefficients of variation of the starting levels for each site/year. (**E**) The coefficients of variation of the single shutdown rates for the different plantings and sites in Sweden in 2014. (**F–G**) Similarly, for Sweden 2016 and Norwich in both years, separating the V-indep rates (**F**) and V-dep (**G**). Data from *Source data 2* and *3*.

hand, there was no significant difference in the variability of the starting levels (*Figure 5D*) between the different field sites and years, and similarly for the shutdown rates (*Figure 5E-G*). What we describe as the starting level was measured after some days in the field, so it is not equivalent to a non-vernalized control. Some *FLC* shutdown, most likely VIN3-independent, will have occurred at that time. Therefore, it seems that the combination of these two determinants (starting levels and rate of VIN3-independent shutdown), which together we call 'autumnal expression', provides most of the variation in *FLC* levels between these accessions and NILs.

## Vernalization is saturated before midwinter in the field

Having characterised how the haplotypes vary in their *FLC* dynamic response to field conditions, we investigated whether this variation had an effect on the floral transition in the field.

In the 2014–5 season, all accessions and NILs bolted relatively synchronously in spring in Norwich and Sweden, although there were small but significant differences in both cases. In Norwich, the flowering time difference between the NILs and Col *FRI* was small, and in Sweden not significant, suggesting that most of the variation seen between the accessions did not derive from the *FLC* haplotype (*Figure 6—figure supplements 1* and *2*). There was also very little difference in branch production in the NILs and accessions in Norwich (*Figure 6—figure supplement 1*), which was surprisingly low, despite the known role of *FLC* in suppressing branching (*Huang et al., 2013*; *Lazaro et al., 2018*). Nevertheless, we confirmed that this effect does occur in the accessions and NILs in controlled condition experiments, in which longer vernalization times resulted in increased numbers of branches (*Figure 1—figure supplement 1*).

The observed flowering times suggested that the vernalization requirement is normally saturated during the winter in the field. To test this hypothesis, in 2016–7 we transferred plants from field conditions to heated, long-day conditions at different stages, and scored the length of time until floral buds were visible at the shoot apex (bolting). There was wide variation and substantial delay to bolting within the genotypes for plants transferred to the warm greenhouses early in autumn, at all sites, indicating that vernalization requirement had not been saturated at this point (*Figure 6A,C,E*). Indeed, in both Swedish sites many plants did not flower within the experimental time (136 days after transfer for South, and 104 days for the North) when transferred 25 days after sowing. Some genotypes (particularly Löv-1) required more than 46 days in the field before most individuals were competent to flower in warm, long-day conditions (*Figure 6—figure supplement 3*). There was also clear variation between genotypes in their time to flowering in the earlier transfers. However, as winter progressed, time to bolt after transfer reduced quantitatively and became more uniform for plants within and between each genotype (*Figure 6B,D,F*), although vernalization saturated at different rates in different genotypes (*Figure 6—figure supplements 3–4*). All accessions and NILs bolted broadly synchronously and rapidly after removal on 21st December in Norwich, in North Sweden NILs and accessions were almost synchronous by 24th November, in line with previous findings (*Duncan et al., 2015*), and in South Sweden FRI-containing lines were broadly synchronous by the last transfer on 17th December (*Figure 6*, *Figure 6—figure supplements 3–4*). Therefore, in current climates almost all Arabidopsis plants have probably saturated their requirement for vernalization before midwinter, ensuring that flowering time is not delayed the following spring.

## Relationship of *FLC* expression and timing of flowering

We tested whether the observed effects on flowering time were linked to the varying levels of *FLC* in transferred plants. Across genotypes, in Norwich and North Sweden 2016–7, the time to the floral transition after transfer from the field correlated closely with *FLC* expression at the time of transfer, as expected (Norwich, $R^2$ = 0.68, p<0.001, North Sweden, $R^2$ = 0.85, p<0.001, linear regression *Figure 6G,H*). Accessions, NILs and mutants with high starting levels and slower downregulation rates generally bolted later and took longer to saturate their vernalization requirement, indicating that the variation in *FLC* dynamics in autumn has a phenotypic effect.

However, each accession differed in the relationship between *FLC* levels at transfer and subsequent bolting time (*Figure 6—figure supplements 5–6*). For example, for Löv-1 and the Löv NIL1, the time to bolt for a given level of *FLC* at transfer is longer than for Col *FRI*, but for Edi-0, it is shorter (*Figure 6—figure supplements 5–6*). This analysis allowed us to extract a further feature of *FLC* regulation (*Figure 2A*). We named this feature the *FLC*-post-vern value (*m* in *Table 1*, days-to-

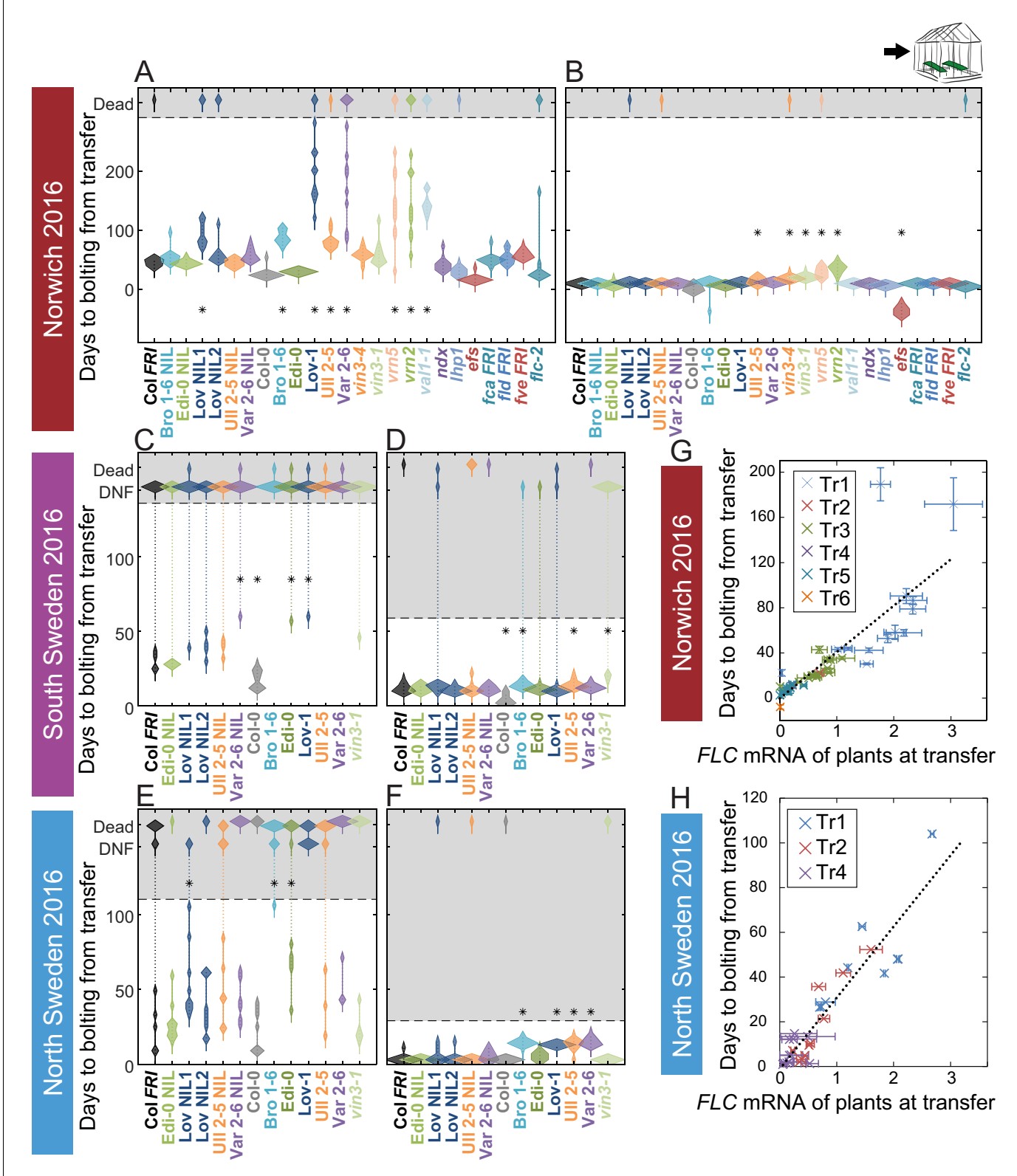

**Figure 6.** Vernalization requirement for *FLC* downregulation is saturated in natural winters. (A–F) Bolting time for accessions and NILs after transfer to floral-induction conditions from 'natural' winter 2016–7, in (**A**) Norwich 21/10/16 (**B**) Norwich 21/12/16 (**C**) South Sweden 01/10/2016 (**D**) South Sweden 17/12/16 (**E**) North Sweden 06/09/2016 (**F**) North Sweden 24/11/2016. Plants that did not flower by 14/02/17 (**C, D**) or 23/12/16 (**E, F**) are shown as DNF and dead plants are indicated. Plots show the histogram of numbers of plants as the width of violin plots. A line connects the measurements to indicate

*Figure 6 continued on next page*

*Figure 6 continued*

the range. Flowering time of genotypes that are significantly different to the reference line Col *FRI* are indicated by * (ANOVA with Dunnett's post-hoc test). p-values for all comparisons are given in *Supplementary file 3*. (G–H) North Sweden 2016 transfers for accessions and NILs, (G) mean time to bolting after transfer to floral-inductive conditions plotted against mean *FLC* expression per genotype at transfer, Norwich 2016–7, linear regression $R^2$ = 0.68, p<0.001. (H) Mean time to bolting after transfer to floral-inductive conditions plotted against mean *FLC* expression per genotype at transfer, North Sweden 2016–7. Genotypes that did not bolt within 205 days not shown, linear regression $R^2$ = 0.85, p<0.001. N = 12 plants except where plants died or (E, H) did not bolt within 205 days (*Source data 5*). Error bars for G and H show s.e.m.

The online version of this article includes the following figure supplement(s) for figure 6:

**Figure supplement 1.** Flowering after winter in Norwich 2014–5 in the field was largely synchronous.
**Figure supplement 2.** The transition to flowering after natural winters in South and North Sweden 2014–5 in the field was largely synchronous, while later bolting had a negative effect on survival only in South Sweden.
**Figure supplement 3.** Bolting after transfer to warm, long-day conditions from winter in the field 2016–7 saturates at different rates in different genotypes in Sweden.
**Figure supplement 4.** Bolting after transfer to warm, long-day conditions from winter in the field 2016–7 saturates at different rates in Norwich.
**Figure supplement 5.** The relationship between time to floral transition and *FLC* expression at the end of cold (Norwich winter 2016–7) varies among accessions, both due to trans effects and due to the *FLC* alleles themselves.
**Figure supplement 6.** The relationship between time to floral transition and *FLC* expression at the end of cold in North Sweden winter 2016–7.
**Figure supplement 7.** Increased vernalization increases the amount and reduces the variability of seed set.

bolting-per-*FLC*-unit), representing the slope of the regression between *FLC* spliced RNA levels at the time of transfer (x-axis) to the subsequent floral transition time for these transferred plants (y-axis, as for *Figure 6G and H*). This means that, for a given level of *FLC* at the time of transfer, a genotype with a higher *m* 'post-vern' value flowers later than one with a lower *m* value.

We calculated this value individually for each genotype from both the Norwich and the North Sweden transfers (*Figure 6—figure supplements 5–6*), using the equation given in the legend for *Table 1*. There was substantial variation between the different glasshouses in the estimates for the accessions (*Table 1*), suggesting that there is a large genotype-by-environment effect on this parameter due to non-*FLC* effects. Moreover, for Col-0 and Löv-1, estimations of *m* were unreliable (p>0.1) due to flowering occurring rapidly even without vernalization (as *FLC* values are too low to cause measurable differences in flowering: Col-0) or not at all (as in North Sweden *FLC* values were

**Table 1.** Linear regression relationship between bolting time and *FLC* mRNA expression, as shown in *Figure 6—figure supplements 5–6*, where *days to bolting = m[FLC mRNA] + c*, and *m* is the '*FLC*-post-vern' value and *c* is the y-intercept fitted constant relating to non-FLC-mediated bolting delay (dependent on the transfer dates and conditions of each experiment).

NA – estimate only based on two data points, so no standard error is calculable. n.d. – no data. No value given for Col-0 as initial *FLC* value is too low for useful estimation.

| Genotype | Norwich m (days-to-bolting-per-*FLC*-unit) | Norwich Std. error | Norwich p-value | North Sweden M | North Sweden Std. error | North Sweden p-value | Average m 'post-vern' |
|---|---|---|---|---|---|---|---|
| Col *FRI* | 36.5 | 5.5 | 0.007 | 39.0 | 17.6 | 0.269 | 37.8 |
| *vin3-1 FRI* | 51.0 | 3.9 | 0.049 | 67.3 | 27.6 | 0.247 | 59.1 |
| Bro NIL | 36.1 | 0.3 | 0.006 | n.d. | n.d. | n.d. | 36.1 |
| Edi NIL | 33.5 | 2.7 | 0.051 | 39.8 | 1.1 | 0.017 | 36.7 |
| Löv NIL1 | 40.7 | 5.0 | 0.079 | n.d. | n.d. | n.d. | 40.7 |
| Löv NIL2 | 27.4 | 0.8 | 0.019 | 23.8 | 0.5 | 0.014 | 25.6 |
| Ull NIL | 24.8 | 1.0 | 0.025 | 42.1 | 9.3 | 0.138 | 33.4 |
| Var NIL | 25.7 | 0.9 | 0.022 | 22.3 | 1.6 | 0.046 | 24.0 |
| Bro1-6 | 27.0 | 0.3 | 0.007 | 37.3 | 2.1 | 0.036 | 32.1 |
| Edi-0 | 16.4 | 3.2 | 0.124 | 48.3 | 9.2 | 0.119 | 32.4 |
| Löv-1 | 113 | 19 | 0.110 | n.d. | n.d. | n.d. | 113 |
| Ull2-5 | 31.4 | 1.2 | 0.024 | 16.9 | 8.5 | 0.297 | 24.2 |
| Var2-6 | 63.1 | 6.9 | 0.070 | 35.0 | NA | NA | 49.0 |

too high to allow flowering and its timing to be measured within this experiment: Löv-1). Practically, therefore, although any genotype with functional *FLC* should have a theoretical post-vern value, it can only be measured for high-*FLC* genotypes under conditions in which they will nevertheless flower.

The *m* post-vern value also varied between the NILs and the common Col *FRI* parent within each site, suggesting that this relationship is also influenced by cis variation at *FLC*. Within the lines in the common Col *FRI* background, the estimates correlated more closely across the two glasshouse environments, with the exception of the Ull NIL. Therefore, within the Col *FRI* background, we can quantify the effect of *FLC* levels on bolting time in warm, long-photoperiod environments, and find that the *FLC* haplotypes have different effects after vernalization, as well as different responses to vernalization. This effect may be linked to reactivation of *FLC* expression in the warm, as the *vin3-1* and Löv-1 NIL1, both of which have reactivation phenotypes (*Sung and Amasino, 2004*; *Duncan et al., 2015*; *Qüesta et al., 2020*), have high 'post-vern' values.

## High autumnal *FLC* reduces precocious flowering in a warm autumn

At all sites in the 2014–5 season, and in South Sweden in 2016–7, across all accessions and NILs, flowering in the field occurred after midwinter (*Figure 6—figure supplements 1–2*, *Figure 7—figure supplement 1*), by which time *FLC* levels were below the level at which they influenced flowering time in the transfers (*Figure 2—figure supplement 1*, *Figure 3—figure supplement 1*, *Figure 6*). Therefore, although we observed natural variation in *FLC* levels in autumn (*Figure 5*), at most sites in most years the conditions were such that this did not result in flowering time variation. However, in the warm North Sweden autumn 2016–7 many of the plants with the Col background (NILs, Col *FRI*, and *vin3-1 FRI*) transitioned to flowering early, by the 18th November, before winter and snowfall (*Figure 7A*). This precocious bolting was rare in the SV accessions and was much reduced even in the RV accession Edi-0. Over all the genotypes, the percentage of plants transitioning to flowering before winter negatively correlated with genotype *FLC* expression on 5th October, one day after the first recorded bolting for plants in the field (p<0.001, Generalised Linear Models (GLMs) for binomial data, *Figure 7B*). This was before substantial upregulation of *VIN3*, indicating that this variation in *FLC* must derive from the combination of the starting levels and VIN3-independent shutdown. Within the Col *FRI*^SF2 background the Var and Löv *FLC* haplotypes, the NILs with the highest *FLC* levels during autumn (*Figure 3—figure supplement 1*), substantially reduced precocious bolting (p=0.005, binomial proportions test, *Figure 7A*). Thus, variation in *FLC* starting expression and shutdown (collectively called autumnal expression) results in variation in the alignment of flowering with the seasons.

## Variation in *FLC* expression correlates with fitness in the field

Across all genotypes, plants that bolted precociously were not observed to set seed before winter. In this year (2016) and at this site (N.Sweden), mortality was high, compared to other years and sites, and plants that bolted precociously were less likely to survive the winter (42% of bolting plants survived, whereas 67% of non-bolting plants survived to spring, p<0.001, binomial proportions test, *Figure 7—figure supplement 2A*). Therefore, high *FLC* levels in October correlated with higher survival to seed set in the field (p<0.003, GLM for binomial data, *Figure 7—figure supplement 2B*).

We therefore investigated whether *FLC* was linked to total genotype fitness, as measured by silique number, and found a complex pattern (*Figure 7C*). Despite many of the NILs having a high level of precocious flowering, several NILs produced more seed than the locally-adapted accessions (*Figure 7C*, *Figure 7—figure supplement 2C*).

To better understand why some NILs had higher fecundity than others, we investigated whether *FLC* might have an additional effect on silique production. In the Col *FRI* background plants, there was no overall penalty in average silique number for individual surviving plants that bolted before winter compared to those that transitioned afterwards (*Figure 7C*, *Figure 7—figure supplement 2D*), suggesting that the differences were not due to an *FLC* effect on flowering time. Saturating vernalization increased seed set in the 2016 Norwich transfer experiments, as plants that were transferred before saturation of vernalization requirement produced lower and more variable amounts of seed (*Figure 6—figure supplement 7*). However, in the field in North Sweden 2017 silique production occurred only after winter, after saturation of vernalization requirement (*Figure 6F*). We noted

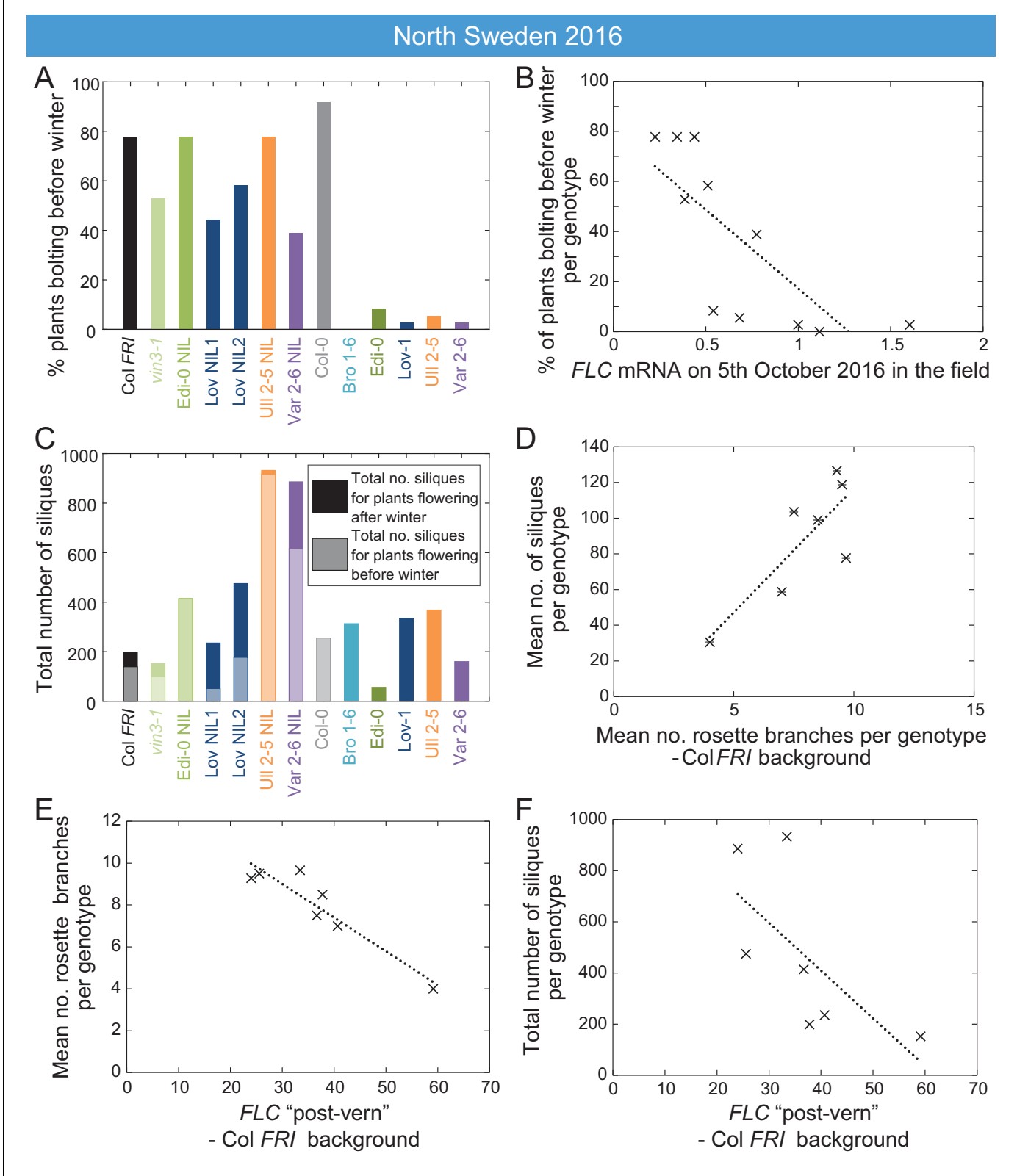

**Figure 7.** High FLC reduces precocious bolting in North Sweden in warm years. (**A**) Percentage of plants bolting before winter in the North Sweden 2016 experiment by genotype. Plants in the field were less likely to flower precociously before winter (18th November 2016) if they are accessions from more northerly latitudes or, to a lesser degree, if they are *FLC* introgression lines containing *FLC* haplotypes from SV accessions. (**B**) The percentage of plants transitioning to flowering before winter per genotype negatively correlated with *FLC* expression (normalised to control) on 5th October

*Figure 7 continued on next page*

*Figure 7 continued*

(R$^2$ = 0.59, p=0.0058). (**C**) Total number of siliques produced per genotype, showing contribution from plants that bolted before winter and plants that bolted after. Within the Col *FRI* genetic background there was no overall penalty in average silique number for surviving plants bolting before vs. after winter (92 and 77 per plant respectively, not significant in Mann-Whitney U test). (**D**) Mean silique production in plants surviving to set seed positively correlated to their mean rosette branch production for Col *FRI* genetic background genotypes (NILs and *vin3-4*; R$^2$ = 0.56, p-value=0.002). (**E**) Rosette branch production of Col *FRI* genotypes surviving to set seed is strongly negatively correlated with the *FLC* post-vern value for that genotype as from *Table 1* (R$^2$ = 0.86, p-value<0.002). (**F**) Total number of siliques produced by Col *FRI* background genotypes plotted against *FLC* post-vern, linear regression for post-vern effect alone, R$^2$ = 0.35, p-value=0.1. N = 36 plants sown (**A–C**), n for surviving plants (**D–F**) varies per genotype, see *Source data 6*.

The online version of this article includes the following figure supplement(s) for figure 7:

**Figure supplement 1.** Flowering in the field across all sites in 2016–2017.

**Figure supplement 2.** *FLC* affects fitness in North Sweden through bolting time and branching.

that there was large variation in branching within the Col *FRI* background plants, and branching is known to be affected by *FLC* (*Figure 1—figure supplements 1–2*; *Huang et al., 2013*; *Lazaro et al., 2018*). We asked whether the likelihood of *FLC* reactivation after winter might affect branching and thereby silique production. To estimate this, we used as proxy for likely *FLC* reactivation the relationship between *FLC* and bolting time in the warm, the '*FLC*-post-vern value' as derived from *Figure 6—figure supplements 5–6* (*Table 1*). This did not seem to affect survival to seed set or date of bolting of the survivors (*Figure 7—figure supplement 2E,F*). Instead, within the Col *FRI* background, the average silique set among survivors correlated with rosette branching (*Figure 7D*, *Figure 7—figure supplement 2G*) and branch number was strongly negatively correlated with *FLC*-post-vern, though not *FLC* expression (*Figure 7E*, *Figure 7—figure supplement 2H*).

As expected, survival was the principle factor explaining total silique production (*Figure 7—figure supplement 2I*). However, we found that although the post-vern value of *FLC* did not well explain total genotype fitness on its own (*Figure 7F*), when combined with survival to seed set both factors explained a large part of the variation in genotype fitness in Col *FRI* background (linear model adjusted R$^2$ = 0.94, p-value=0.001 for model, post-vern p-value=0.006, percentage survival to silique set p-value=0.002, *Figure 7F*, *Figure 7—figure supplement 2I*).

## Discussion

Investigation of the dynamics of key floral integrators in the field has recently led to important insights into their molecular response to natural environments (*Antoniou-Kourounioti et al., 2018*; *Song et al., 2018*). Here, we integrated experiments on the expression of *FLC* in field conditions with investigation on how natural genetic variation and induced mutation interact in distinct climates to affect phenotype and fitness. In our previous work, we found that, as expected from laboratory studies, *FLC* was repressed and *VIN3* was induced by winter cold (*Antoniou-Kourounioti et al., 2018*; *Hepworth et al., 2018*). However, in 2014–5, winter conditions in Norwich generated a subtly different response to those in Sweden, activating the VIN3-independent transcriptional shutdown before the VIN3-dependent epigenetic pathway (delay in *VIN3* onset and slow then fast shutdown of *FLC* in Norwich, *Figure 2—figure supplement 1*). In the following season, this pattern was triggered again, not only in Norwich, but also in North Sweden, implying that this is a common occurrence that plants must adjust to across climates, although with different frequencies at different locations (*Figure 3—figure supplement 1*).

We find that variation in *FLC* mRNA levels in both accessions and NILs is largely due to differences in both the starting *FLC* level and the VIN3-independent rate of shutdown (collectively called autumnal expression), which vary widely between accessions but not between field sites or years (*Figure 5*). This variation in rate of response, as well as absolute levels of *FLC*, partly explains why *FLC* levels measured at any one time may not correlate with the final flowering time phenotype in different accessions (*Sasaki et al., 2018*).

In Norwich, we investigated the mechanisms of the responses. Mutants of the autonomous pathway, which increase *FLC* expression in *fri* plants in laboratory conditions (*Ausín et al., 2004*; *Liu et al., 2010*; *Wu et al., 2016*), generally behaved in a similar manner to Col *FRI* in terms of vernalization response (*Figure 4*). However, the reduced response of the *fve* mutants in the VIN3-

independent phase, though not significant in each year, may be worth further investigation, as FVE has been implicated in intermittent cold-sensing through histone deacetylation at *FLC*, independently of vernalization (*Kim et al., 2004*; *Jung et al., 2013*).

All three factors that set *FLC* levels in the field (starting levels, VIN3-independent and VIN3-dependent phases) require chromatin modifiers for their correct function. The VIN3-independent phase involves chromatin-modifier regulation of transcription, with the VIN3-dependent phase using a similar set of modifiers. These chromatin regulators are rarely found in genome-wide-association studies for flowering time, likely because their alteration would have pleiotropic effects. In the natural accessions we find that the VIN3-dependent (epigenetic) phase is indeed the least variable (*Figure 5*). This may explain why so much variation in vernalization maps to *FLC* itself (*Lempe et al., 2005*; *Sánchez-Bermejo et al., 2012*; *Dittmar et al., 2014*; *Sasaki et al., 2015*; *Bloomer and Dean, 2017*; *Sasaki et al., 2018*). Nevertheless, we did observe variation in *VIN3* expression in Var2-6, which seems to derive from differences in circadian regulation (*Figure 3—figure supplement 3*), and QTLs over the *VIN3* region have previously been found to affect flowering (*Dittmar et al., 2014*; *Méndez-Vigo et al., 2016*; *Ågren et al., 2017*). Variation in *VIN3* expression in the other accessions we tested was more complex than the general downregulation in Var2-6. Given the intricate regulation of *VIN3* induction (*Antoniou-Kourounioti et al., 2018*), identifying the source of this difference will require detailed investigation.

The importance of *FLC* variation in adaptation for natural populations is well established (*Méndez-Vigo et al., 2011*; *Sánchez-Bermejo et al., 2012*; *Dittmar et al., 2014*; *Li et al., 2014*; *Duncan et al., 2015*; *Ågren et al., 2017*; *Bloomer and Dean, 2017*). However, we found that even in a challengingly warm year, vernalization requirement saturated before midwinter across three climates (*Figure 6*). This response had been seen previously in North Sweden in the locally-adapted Löv-1 accession as a reaction to the extreme winters (*Duncan et al., 2015*). However, we find that this is a general response in all our tested accessions. The principle exceptions were the *vin3* and *vrn* mutants, which had notably delayed flowering in both years at the Norwich site. The consequences of incomplete vernalization are severe – in our 2016–7 transfer experiments this led to delayed flowering and reduced fecundity (*Figure 6*, *Figure 6—figure supplement 7*), and several field experiments have found a general trend towards earlier flowering promoting fitness in environments with warmer winters (*Ågrena et al., 2013*; *Fournier-Level et al., 2013*; *Grillo et al., 2013*; *Dittmar et al., 2014*; *Ågren et al., 2017*; *Exposito-Alonso et al., 2018*). In 2014–5, the high synchronicity of flowering in all the accessions and NILs after winter and the fact that flowering aligned with spring across our field sites (*Figure 6—figure supplement 2*), suggests that in most plants vernalization has adapted to reduce the delaying effect of *FLC* on flowering as early as possible.

Nevertheless, we observed that the Swedish, cold-winter-adapted accessions and *FLC* haplotypes associated with slow vernalization expressed the highest *FLC* levels throughout autumn and early winter (*Figures 2* and *3*, *Figure 2—figure supplement 1*, *Figure 3—figure supplement 1*). In North Sweden in 2016–7, we captured behaviour for a warmer growing season that revealed an adaptive role for these high *FLC* expression levels. There was a relatively warm September that year with a long period with less precipitation (*Swedish Meteorological and Hydrological Institute, 2020*, www.smhi.se/data/meteorologi/ladda-ner-meteorologiska-observationer), which required us to water the plants to avoid drought stress. This, in combination with the earlier sowing (warmer and longer days) meant that the equivalent period of growing time for the plants (relative to their germination time) was more inductive to growth and flowering. This allowed us to capture a different behaviour. Plants are exposed to both types of the years we recorded, since the timing of both sowing dates was within the natural germination range, although in natural environments seed dormancy has a strong influence on phenology and is highly responsive to temperature and microclimate, and affected by *FLC* itself (*Springthorpe and Penfield, 2015*; *Kerdaffrec et al., 2016*; *Marcer et al., 2018*). In our experiments, in plants with a haplotype characteristic of a slow-vernalizing accession, including the locally-adapted Swedish accessions, the higher level of *FLC* expression protected against precocious flowering and its consequent reduced survival (*Figure 7*, *Figure 7—figure supplement 2*). These rare, but highly selective occurrences, may play a role in local adaptation, although we do not know the reason behind the high mortality in this year. However, that the effects of these haplotypes are only revealed occasionally is a logical consequence of the fact that the flowering time genes can have strong or weak effects depending on the environment (*Wilczek et al.,*

*2009*; *Sasaki et al., 2015*; *Burghardt et al., 2016*; *Fournier-Level et al., 2016*; *Sasaki et al., 2018*; *Taylor et al., 2019*).

*FLC* also controls fecundity as well as survival (*Ågrena et al., 2013*). *Li et al., 2014* found that *FLC* haplotypes from SV accessions produced lower seed weight compared to those from RV accessions in non-saturating vernalization conditions, though the mechanism for this effect is unknown. In the field, silique production is closely linked to branch production (*Figure 7*; *Taylor et al., 2019*). In Arabidopsis, saturation of vernalization requirement is known to increase flowering branch production, particularly rosette branch production, and this effect is linked to *FLC* (*Figure 1—figure supplements 1–2*; *Huang et al., 2013*; *de Jong et al., 2019*). As expected, *FLC* levels in autumn did not correlate with branch production in spring (*Figure 7—figure supplement 2H*), and in Norwich 2014–5, there was little variation in branch production. However, we found that under the conditions in the field in North Sweden, branch production was negatively related to a factor we named '*FLC* post-vern', which encoded the relationship between *FLC* levels and subsequent flowering in warm controlled conditions (*Figure 7*). Variation in this factor probably derives from regulatory differences at *FLC* post-cold, such as the reactivation phenotype of the Löv-1 *FLC* allele (*Shindo et al., 2006*; *Coustham et al., 2012*; *Duncan et al., 2015*; *Qüesta et al., 2020*), allowing *FLC* repression of flowering to saturate at the shoot apical and axillary meristems at different rates, a phenomenon that occurs in the perennial relative *Arabis alpina* (*Wang et al., 2009*; *Lazaro et al., 2018*). Why the field conditions in Sweden, but not Norwich, revealed this effect is not yet clear.

In summary, our detailed analysis of the different phases of *FLC* silencing through winters in distinct climates, over multiple years, has shed light on adaptive aspects of the vernalization mechanism. Autumnal *FLC* expression, made up from *FLC* starting levels and early phases of *FLC* silencing, is the major determinant for variation in *FLC* levels during vernalization (*Figure 5*). *FLC* downregulation aligns flowering response to spring and non-coding variation affects this alignment in different climatic conditions and year-on-year fluctuations in natural temperature profiles. In a changing climate, understanding the complex genotype-by-environment interactions that govern timing mechanisms will become ever more important.

## Materials and methods

### Plant materials
Sources of previously described mutant lines and transgenics are presented in *Supplementary file 4*. Requests for materials should be addressed to Caroline Dean.

#### NILs
All near-isogenic lines were produced by six rounds of backcrossing to the Col *FRI* parent, selecting for the introgressed *FLC* in each generation, before one round of selfing and selection of homozygous families.

### Experimental conditions
#### Field experiments
Field experiments have been described previously (*Antoniou-Kourounioti et al., 2018* (2016-7 winter); *Hepworth et al., 2018* (2014-5 winter)). Briefly, seeds were stratified at 4°C for three days. For gene expression measurements, for all field sites and sowing dates and timepoints within them, six replicate tray-cells were sown using a block-randomised design within 5.7 cm 28 cell trays (Pöppelman, Lohne, Germany), and where each replicate included material from at least three plants. For flowering time, plants were thinned to a single plant per cell in 3.9 cm 66 cell trays (Pöppelman). Trays were watered when necessary.

In Norwich, trays were placed on benches in an unheated, unlit glasshouse, and bedded in vermiculite. For expression, plants were randomised within six single-replicate sample-sets, which were then randomised using Research Randomiser (*Urbaniak and Plous, 2015*) in a three complete block design lengthwise along the greenhouse, adjusting to ensure each of the two replicates per block were on different benches. For flowering time, plants were block-randomised per tray and per block.

In Sweden, trays were germinated and grown outside under plastic covers for two weeks at Mid Sweden University, Sundsvall (North Sweden) or Lund University (South Sweden), due to the difficulty of sowing directly into the field and to ensure sufficient germination for analysis, based on previous experiments. Trays were then moved to the experiment sites and dug into the soil. The experiment site in the North was at Ramsta (62° 50.988′N, 18° 11.570′E), and in the South at Ullstorp (56° 06.6721′N, 13° 94.4655′E). Expression sample-sets were randomised in three blocks. Flowering time plants were in a completely randomised design across two (2014–5) or three (2016–7) blocks.

Plants were sown and moved to the field site on: Norwich first year, sown into position on 29th September 2014, second year, sown into position on 15th September 2016: South Sweden first year, sown 24th September 2014, moved 8th October 2014, second year sown on 6th September 2016, moved on 21st September 2016: North Sweden first year, first planting 'A' sown 26th August 2014, moved 11th September 2014; second planting 'B' sown 8th September 2014, moved 24th September 2014: North Sweden second year, sown on 12th August 2016, and moved 24th August 2016.

For the transferred plants, in Norwich 2016–7, for each transfer six trays (each holding two replicates, total N = 12) were moved from the unheated, unlit, ventilated greenhouse to a greenhouse with supplementary lighting (600W HPS lamps) and heating set to 22°C/18°C, 16 light/8 hr dark, and 70% humidity on 21st October 2016 (22 days after sowing), 3rd November 2016 (35 das), 17th November 2016 (49 das), 30th November 2016 (62 das), 21st December 2016 (83 das) and 26th January 2017 (119 das). For selected time points, plants were covered with ventilated clear plastic bags to collect seed for weighing. In South Sweden, trays were transferred from the field site to heated, lit greenhouses at Lund University, on 1st October (25 das), 22nd October (46 das), 19th November (74 das) and 17th December 2016 (102 das). In North Sweden, for each transfer three trays (each holding four replicates, N = 12) were moved from the field site to a greenhouse set to 16 hr light, 22°C, at Mid Sweden University, Sundsvall (as in *Duncan et al., 2015*), on 6th September (25 das), 4th October (53 das), 1st November (81 das) and 24th November 2016 (104 das).

For all expression analysis except the 48 hr sampling (*Figure 3—figure supplement 3*), six replicates per timepoint per genotype were chosen in order to allow sufficient number of samples for statistical analysis while allowing for losses in the field and to allow duplication within randomisation blocks. Where resulting samples are smaller, this is due to experimental or processing loss (e.g., death of plants in the field, degradation due to poor sample quality or processing, see RNA extraction and QPCR). For *Figure 3—figure supplement 3 and*   replicates per timepoint per genotype were chosen due to space constraints. Each expression sample (single replicate) was of at least three plants pooled. For flowering time, 12 plants per genotype per transfer condition were chosen to provide replication across blocks and trays while remaining within size constraints. For field flowering in North Sweden 2016–7, 36 plants per genotype were sown to allow for losses, although in the event these were more substantial than anticipated.

Temperature was recorded at plant level at each site with TinyTag Plus two dataloggers (Gemini Data Loggers (UK) Ltd, Chichester, UK). Bolting was scored when flower buds were visible at the shoot apical meristem. For the North Sweden 2016–7 field experiment, plants were scored for survival and flowering in the field from planting to December 2016, and then from March to May 2017. Plants were harvested and scored for branching and silique production after the end of flowering, in July 2017.

## Branching analysis

; Seeds were sown on soil, stratified after sowing for three days at ~4–5°C, and transferred to a Norwich long-day glasshouse set to 18°C/15°C, 16/8 hr light/dark conditions for 7 days before being returned to vernalization conditions (a 4°C growth chamber under short day, low light conditions; 8/16 hr light/dark) for 12, 8, 4, and 0 weeks. Sowing was staggered so that after vernalization all plants were transferred to glasshouse conditions simultaneously. Plants were scored for their flowering time, total branch number, cauline branch number and rosette branch number. In all cases, plants were randomised into blocks and at least three replicate plants for each accession/cultivar per treatment were scored for their flowering and branching phenotypes. Primary rosette and cauline branch number were scored at senescence.

## RNA extraction and QPCR

RNA extraction and QPCR for field experiments were performed as described in *Hepworth et al., 2018* and *Antoniou-Kourounioti et al., 2018*. Field data were unified across sites and timepoints within the yearly datasets and normalised to a synthetic control sample, as described in *Hepworth et al., 2018*, to give the 'normalised concentration' reported in the results. QPCR results were analysed using LinReg (*Ruijter et al., 2009*), and normalised to the geometric means of At5g25760 ('*PP2A*') and At1g13320 ('*UBC*') control genes (*Czechowski et al., 2005*; *Yang et al., 2014*).

QPCR samples that showed high Cp values (UBC Cp >28 for LinReg analysis) of the control genes, indicating possible degradation, were excluded if test amplicon (FLC, VIN3) results were also anomalous (criteria: absent, or varying from non-flagged samples by an order of magnitude). Any measurements where amplicon Cp values varied by more than 0.6 were also excluded if insufficient sample was available for a repeat.

Primers used are described in *Supplementary file 5*.

## Statistics

The rates of *FLC* shutdown were estimated using linear regression, as shown in *Figure 2A*. For Norwich 2014–5 and for all sites in 2016–7, a separate rate was calculated for timepoints before the point of induction of *VIN3* (Norwich 2014: ~58 days, Norwich 2016: ~48 days, South Sweden 2016: ~35 days, North Sweden: ~46 days) and for those after. For North and South Sweden 2014–5, when the *VIN3* induction was not delayed, all timepoints were combined, excluding measurements after 155 days. These timepoints were during or after the snow, when temperatures started to increase and reactivation was observed in some genotypes.

For the analysis presented in *Supplementary file 1* and *Figures 2–4*, the Starting level comparisons and the Shutdown rate comparisons were done separately, using the following methods. For the starting levels, ANOVA was performed in the R (*R Development Core Team, 2018*) statistical language using the lm function, followed by a Dunnett post-hoc test comparing all genotypes against the control Col *FRI*. For the shutdown rates, the lmer function from the lmerTest package in R was used to perform the same comparison on the genotype-timepoint interaction, controlling for blocks as random factors and after setting the mean of the timepoints to 0 on the x-axis, using the default Satterthwaite's method for the post-hoc t-tests.

We used the R package cvequality (Version 0.1.3; *Marwick and Krishnamoorthy, 2019*) with the asymptotic test (*Feltz and Miller, 1996*) to assess differences between the coefficients of variation of different groups as described in the text and *Supplementary file 2*. The significance limit was adjusted to control the false discovery rate using the Benjamini-Hochberg procedure with a false discovery rate of 0.05 (*Supplementary file 2*, *Supplementary file 6*). This analysis was performed including all the natural accession and NILs combined (*Figure 4*), but also for all the accessions separately and for all the NILs separately (including Col *FRI* in both cases, but not Col-0 or mutants). The different analyses do not change our overall conclusions and all three are reported in *Supplementary file 2*.

Statistical analysis for flowering and branching data (*Supplementary file 3*, *Figure 6*, *Figure 6— figure supplements 1*, *2*, *3* and *4* and *Figure 7—figure supplement 1*) were performed on the plants that flowered during the experiment using the same procedure as for the Starting levels described above.

For multiple regression on field data, R was used to obtain minimal adequate models using linear regression (lm function), except when n > 10 for count data, for which general linear models (GLM, glm function) using Poisson error distributions were used (total numbers of siliques), or for proportion data for which GLMs with binomial errors were used (survival, bolting before winter).

## Acknowledgements

For genetic materials, we are indebted to Prof. Johanna Schmitt (University of California Davis) for the kind gift of *vin3-1 FRI*, and Huamei Wang for creating the Ull and Bro NIL lines and preparing seed stocks. For the Swedish field sites, many thanks to family Öhman and Nils Jönsson. Thanks to Prof. James Brown (JIC) for advice on the statistics, and to Ian Baldwin (Senior Editor at eLIFE) for

helpful suggestions on the initial submission of the manuscript. Finally the authors would like to reiterate their appreciation of all those from the Dean, Howard, Holm, Säll and Irwin groups who helped in cold, heat, wind, rain and laboratory with the field studies, plus Ingalill Thorsell at Drakamöllan Gårdshotell for making the lab retreats during the field studies so enjoyable.

## Additional information

### Funding

| Funder | Grant reference number | Author |
|---|---|---|
| Horizon 2020 Framework Programme | MEXTIM | Jo Hepworth<br>Rea L Antoniou-Kourounioti<br>Kristina Berggren<br>Catja Selga<br>Eleri H Tudor<br>Deborah Cox<br>Barley Rose Collier Harris<br>Judith A Irwin<br>Martin Howard<br>Torbjörn Säll<br>Svante Holm<br>Caroline Dean |
| Biotechnology and Biological Sciences Research Council | BB/J004588/1 | Jo Hepworth<br>Rea L Antoniou-Kourounioti<br>Eleri H Tudor<br>Deborah Cox<br>Barley Rose Collier Harris<br>Judith A Irwin<br>Martin Howard<br>Caroline Dean |
| Biotechnology and Biological Sciences Research Council | BB/P013511/1 | Jo Hepworth<br>Rea L Antoniou-Kourounioti<br>Eleri H Tudor<br>Bryony Yates<br>Deborah Cox<br>Barley Rose Collier Harris<br>Judith A Irwin<br>Martin Howard<br>Caroline Dean |
| Biotechnology and Biological Sciences Research Council | BB/P003095/1 | Jo Hepworth<br>Eleri H Tudor<br>Judith A Irwin |
| Biotechnology and Biological Sciences Research Council | BB/L016079/1 | Eleri H Tudor |

The funders had no role in study design, data collection and interpretation, or the decision to submit the work for publication.

### Author contributions

Jo Hepworth, Data curation, Formal analysis, Supervision, Investigation, Visualization, Methodology, Writing - original draft; Rea L Antoniou-Kourounioti, Data curation, Formal analysis, Supervision, Investigation, Visualization, Writing - original draft; Kristina Berggren, Catja Selga, Bryony Yates, Deborah Cox, Barley Rose Collier Harris, Investigation; Eleri H Tudor, Formal analysis, Investigation, Methodology; Judith A Irwin, Conceptualization, Supervision, Funding acquisition, Methodology; Martin Howard, Conceptualization, Supervision, Funding acquisition, Methodology, Writing - review and editing; Torbjörn Säll, Svante Holm, Conceptualization, Supervision, Funding acquisition, Investigation, Methodology, Writing - review and editing; Caroline Dean, Conceptualization, Resources, Supervision, Funding acquisition, Investigation, Methodology, Project administration, Writing - review and editing

**Author ORCIDs**

Jo Hepworth (iD) https://orcid.org/0000-0002-4621-8414

Rea L Antoniou-Kourounioti (iD) https://orcid.org/0000-0001-5226-521X

Kristina Berggren (iD) http://orcid.org/0000-0002-7859-9928

Catja Selga (iD) http://orcid.org/0000-0001-8683-1291

Barley Rose Collier Harris (iD) http://orcid.org/0000-0001-5745-1812

Martin Howard (iD) http://orcid.org/0000-0001-7670-0781

Caroline Dean (iD) https://orcid.org/0000-0002-6555-3525

**Decision letter and Author response**

Decision letter https://doi.org/10.7554/eLife.57671.sa1

Author response https://doi.org/10.7554/eLife.57671.sa2

---

## Additional files

### Supplementary files

• Source data 1. Field temperatures.

• Source data 2. RNA Expression for all field experiments 2014–15.

• Source data 3. RNA Expression for all field experiments 2016–17.

• Source data 4. Flowering time and phenotypes for all field experiments 2014–15.

• Source data 5. Flowering time and phenotypes for all transfer experiments 2016–17.

• Source data 6. Flowering time. Survival and phenotypes for field experiments 2016–17.

• Source data 7. Flowering time and branching for accessions and NILs in constant-condition vernalization treatments.

• Supplementary file 1. Statistics for *Figures 2*, *3* and *4*.

• Supplementary file 2. List of all comparisons of coefficients of variation. Differences between the coefficients of variation of different groups (sites, features, years; 24 comparisons)

• Supplementary file 3. Statistics for *Figure 6*, *Figure 6—figure supplements 1–4*, and *Figure 7—figure supplement 1*.

• Supplementary file 4. Sources of previously published mutants and transgenics.

• Supplementary file 5. Primers used for PCR.

• Supplementary file 6. Statistics for *Figure 5*.

• Transparent reporting form

### Data availability

All data generated or analysed during this study are included in the manuscript and supporting files. Source data files have been provided for all figures.

---

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

# Appendix 1

**Appendix 1—key resources table**

| Reagent type (species) or resource | Designation | Source or reference | Identifiers | Additional information |
|---|---|---|---|---|
| Gene (*Arabidopsis thaliana*) | FLOWERING LOCUS C; FLC | TAIR | At5g10140 | |
| Gene (*Arabidopsis thaliana*) | VERNALIZATION INSENSITIVE3; VIN3 | TAIR | At5g57380 | |
| Gene (*Arabidopsis thaliana*) | PROTEIN PHOSPHATASE 2A SUBUNIT A3; PP2A | TAIR | At1g13320 | |
| Gene (*Arabidopsis thaliana*) | UBC; PEX4 PEROXIN4 | TAIR | At5g25760 | |
| Gene (*Arabidopsis thaliana*) | CIRCADIAN CLOCK ASSOCIATED1; CCA1 | TAIR | At2g46830 | |
| Strain, strain background (*Arabidopsis thaliana*) | Col-0 | *Shindo et al., 2005* Doi:10.1104/pp. 105.061309 | Nottingham Arabidopsis Stock Centre (NASC) ID N22625 | |
| Strain, strain background (*Arabidopsis thaliana*) | Bro1-6 | *Long et al., 2013*, Doi:10.1038/ng. 2678 | NASC ID N76726 | |
| Strain, strain background (*Arabidopsis thaliana*) | Edi-0 | *Shindo et al., 2005* Doi:10.1104/pp. 105.061309 | NASC ID N22657 | |
| Strain, strain background (*Arabidopsis thaliana*) | Löv-1 | *Shindo et al., 2005* Doi:10.1104/pp. 105.061309 | NASC ID N22574 | |
| Strain, strain background (*Arabidopsis thaliana*) | Ull2-5 | *Shindo et al., 2005* Doi:10.1104/pp. 105.061309 | NASC ID N22586 | |
| Strain, strain background (*Arabidopsis thaliana*) | Var2-6 | *Shindo et al., 2005* Doi:10.1104/pp. 105.061309 | NASC ID N22581 | |
| Genetic reagent (*A. thaliana*) | Col *FRI*[SF2] | *Lee and Amasino, 1995* Doi:10.1038/ ncomms3186 | | |
| Genetic reagent (*A. thaliana*) | Bro NIL | This paper | | *FLC* from Bro1-5 backcrossed to Col *FRI* background six times and brought to homozygosity, as described in Plant materials section of Materials and methods. Requests for materials should be addressed to Caroline Dean. |

*Continued on next page*

*Appendix 1—key resources table continued*

| Reagent type (species) or resource | Designation | Source or reference | Identifiers | Additional information |
|---|---|---|---|---|
| Genetic reagent (*A. thaliana*) | Edi NIL | This paper | | *FLC* from Edi-0 backcrossed to Col *FRI* background six times and brought to homozygosity, as described in Plant materials section of Materials and methods. Requests for materials should be addressed to Caroline Dean. |
| Genetic reagent (*A. thaliana*) | Löv NIL1 | *Duncan et al., 2015* Doi:10.7554/eLife.06620 | | |
| Genetic reagent (*A. thaliana*) | Löv NIL2 | *Duncan et al., 2015* Doi:10.7554/eLife.06620 | | |
| Genetic reagent (*A. thaliana*) | Üll NIL | This paper, derived from *Strange et al., 2011* doi:10.1371/journal.pone.0019949 | | *FLC* from Üll2-5 backcrossed to Col *FRI* background six times and brought to homozygosity, as described in Plant materials section of Materials and methods. Requests for materials should be addressed to Caroline Dean. |
| Genetic reagent (*A. thaliana*) | Var NIL | *Li et al., 2015* doi:10.1101/gad.258814.115 | | |
| Genetic reagent (*A. thaliana*) | vin3-1 FRI | *Sung and Amasino, 2004* doi:10.1038/nature02195 | | |
| Genetic reagent (*A. thaliana*) | vin3-4 FRI | *Bond et al., 2009b* doi:10.1111/j.1365-313X.2009.03891.x | | |
| Genetic reagent (*A. thaliana*) | vrn1-4 FRI | *Sung and Amasino, 2004* doi:10.1038/nature02195 | | |
| Genetic reagent (*A. thaliana*) | vrn2-1 FRI | *Yang et al., 2017* doi:10.1126/science.aan1121 | | |
| Genetic reagent (*A. thaliana*) | vrn5-8 FRI | *Greb et al., 2007* doi:10.1016/j.cub.2006.11.052 | | |
| Genetic reagent (*A. thaliana*) | ndx1-1 FRI | *Sun et al., 2013* doi:10.1126/science.1234848 | | |
| Genetic reagent (*A. thaliana*) | fca-9 | *Liu et al., 2007* doi:10.1016/j.molcel.2007.10.018 | | |
| Genetic reagent (*A. thaliana*) | fld-4 | *Liu et al., 2007* doi:10.1016/j.molcel.2007.10.018 | | |
| Genetic reagent (*A. thaliana*) | fve-3 | *Ausín et al., 2004* doi:10.1038/ng1295 | | |

*Continued on next page*

*Appendix 1—key resources table continued*

| Reagent type (species) or resource | Designation | Source or reference | Identifiers | Additional information |
|---|---|---|---|---|
| Genetic reagent (*A. thaliana*) | *fca-9 FRI* | This paper | | Cross Col *FRI* and lines reported above. Requests for materials should be addressed to Caroline Dean. |
| Genetic reagent (*A. thaliana*) | *fld-4 FRI* | This paper | | Cross Col *FRI* and lines reported above. Requests for materials should be addressed to Caroline Dean. |
| Genetic reagent (*A. thaliana*) | *fve-3 FRI* | This paper | | Cross Col *FRI* and lines reported above. Requests for materials should be addressed to Caroline Dean. |
| Genetic reagent (*A. thaliana*) | *val1-2 FRI* | *Qüesta et al., 2016* doi:10.1126/science.aaf7354 | | |
| Genetic reagent (*A. thaliana*) | *sdg8 FRI* | *Yang et al., 2014* doi:10.1016/j.cub.2014.06.047 | | |
| Genetic reagent (*A. thaliana*) | *lhp1-3 FRI* | *Mylne et al., 2006* doi:10.1073/pnas.0507427103 | | |
| Sequence-based reagent | UBC_qPCR_F | *Hepworth et al., 2018* doi:10.1038/s41467-018-03065-7 | | CTGCGACTCAGGGAATCTTCTAA |
| Sequence-based reagent | UBC_qPCR_R | *Hepworth et al., 2018* doi:10.1038/s41467-018-03065-7 | | TTGTGCCATTGAATTGAACCC |
| Sequence-based reagent | PP2A QPCR F2 | *Hepworth et al., 2018* doi:10.1038/s41467-018-03065-7 | | ACTGCATCTAAAGACAGAGTTCC |
| Sequence-based reagent | PP2A QPCR R2 | *Hepworth et al., 2018* doi:10.1038/s41467-018-03065-7 | | CCAAGCATGGCCGTATCATGT |
| Sequence-based reagent | FLC_4265_F (spliced sense) | *Hepworth et al., 2018* doi:10.1038/s41467-018-03065-7 | | AGCCAAGAAGACCGAACTCA |
| Sequence-based reagent | FLC_5683_R (spliced sense) | *Hepworth et al., 2018* doi:10.1038/s41467-018-03065-7 | | TTTGTCCAGCAGGTGACATC |
| Sequence-based reagent | FLC_3966_F (unspliced sense) | *Hepworth et al., 2018* doi:10.1038/s41467-018-03065-7 | | CGCAATTTTCATAGCCCTTG |

*Continued on next page*

*Appendix 1—key resources table continued*

| Reagent type (species) or resource | Designation | Source or reference | Identifiers | Additional information |
|---|---|---|---|---|
| Sequence-based reagent | FLC_4135_R (unspliced sense) | *Hepworth et al., 2018* doi:10.1038/ s41467-018-03065-7 | | CTTTGTAATCAAAGGTGGAGAGC |
| Sequence-based reagent | FLC unspliced RT (4029) | *Hepworth et al., 2018* doi:10.1038/ s41467-018-03065-7 | | TGACATTTGATCCCACAAGC |
| Sequence-based reagent | VIN3 qPCR 1 F | *Hepworth et al., 2018* doi:10.1038/ s41467-018-03065-7 | | TGCTTGTGGATCGTCTTGTCA |
| Sequence-based reagent | VIN3 qPCR 1 R | *Hepworth et al., 2018* doi:10.1038/ s41467-018-03065-7 | | TTCTCCAGCATCCGAGCAAG |
| Sequence-based reagent | JF118-CCA1-F | *MacGregor et al., 2013* doi:10.1105/tpc. 113.114959 | | CTGTGTCTGACGAGGGTCGAA |
| Sequence-based reagent | JF119-CCA1-R | *MacGregor et al., 2013* doi:10.1105/tpc. 113.114959 | | ATATGTAAAACTTTGCGGCAATACCT |
| Commercial assay or kit | Turbo DNA Free Kit | Invitrogen | | |
| Commercial assay or kit | SuperScript Reverse Transcriptase III | Invitrogen | | |
| Commercial assay or kit | Roche | LightCycler 480 SYBR Green I Master | | |
| Software, algorithm | R | *R Development Core Team, 2018* | | |
| Software, algorithm | LinReg PCR | https://www. medischebiologie. nl/files/ *Ruijter et al., 2009*, doi:10.1093/nar/ gkp045 | | |

