## [Decision Letter]

**Acceptance summary:**

*FLOWERING LOCUS C* (*FLC*), a MADS-box transcription factor, plays a major role in determining flowering time in *Arabidopsis* in response to vernalization. In this manuscript, the authors analysed *FLC* silencing in natural variants throughout winter in three field sites over multiple years, and revealed that *FLC* starting levels and early phases of its silencing are the major determinants underlying the vernalization response. These findings suggest that the initial expression of *FLC* in natural variants is important for the adaption of vernalization response to natural fluctuating temperature profiles.

**Decision letter after peer review:**

Thank you for submitting your article "Natural variation in autumn *FLC* levels, rather than epigenetic silencing, aligns vernalization to different climates" for consideration by *eLife*. Your article has been reviewed by three peer reviewers, and the evaluation has been overseen by Hao Yu as the Reviewing Editor and Christian Hardtke as the Senior Editor. The following individuals involved in review of your submission have agreed to reveal their identity: Sureshkumar Balasubramanian (Reviewer #1); Ilha Lee (Reviewer #2); Xavier Picó (Reviewer #3).

The reviewers have discussed the reviews with one another and the Reviewing Editor has drafted this decision to help you prepare a revised submission.

Summary:

In this manuscript, the authors suggest that *FLC* starting levels and early phases of its silencing are the major variables underlying the vernalization response via analysing *FLC* silencing in natural variants throughout winter in three field sites (Norwich, Northern Sweden, Southern Sweden).

Essential revisions:

While there was a common interest about the topic and the relevant findings, some major concerns on the description of methodologies and the interpretation of results as listed below need to be addressed to support the conclusion.

1) The Introduction could be further improved with more attractive description of the background and a working hypothesis to be uniquely tested by expression data from field settings in this study.

2) The definition of "aligning vernalization response to different climatic conditions" is unclear. Clarification of the relevant phenotypes, the measures, and the parameters used for this analysis is necessary for understanding the conclusion drawn from this analysis.

3) The meaning of *FLC*-post-vern value (m) in Table 1 is confusing and should be clarified as understanding how this value is calculated and used in this study will considerably influence interpretation of the relevant results. In addition, how the constant related to non-*FLC*-mediated bolting delay is defined also should be explained.

4) The authors should explain the reasons of different sowing and moving (to field sites) schedules for each experiment, and discuss the potential problems pertaining to the results and conclusion that may result from such variability.

5) The authors should clarify with concrete evidence or correct some specific terms, such as "unusual years" and "climate", used in the manuscript.

6) The conclusion and evolutionary implications of this study are not fully supported by the results based on the analysis of existing fitness components and their relationship with the major variable (*FLC* starting levels). The author should re-evaluate whether the traits (survival and fecundity) that were used as the major fitness components are appropriate for this study, and then correct some relevant descriptions and infer the conclusion and the relevant evolutionary implications accordingly.

7) As suggested by all reviewers, the quality of figures and tables as well as their citations in the text should be improved for style and clarity.

Please take into consideration the specific comments from the reviewers below to revise the manuscript.

Reviewer #1:

The manuscript by Hepworth et al., assesses *FLC* expression and its rate of decay and flowering time in a set of genotypes at multiple field sites (Norwich, Northern Sweden, Southern Sweden) in multiple seasons and tries to understand which part of the variable could explain vernalisation response better in natural conditions. The authors use Col FRI in Norwich condition to show that there are 3 aspects of the regulation of *FLC* expression. Initial levels, initial decay in *FLC* expression prior to the induction of *VIN3* and later stages of decay in *FLC* expression that is *VIN3* dependent. The authors argue that primarily the initial levels of *FLC* "aligns vernalization to different climatic conditions". It is nice that the authors use a simple descriptive approach to tease apart some complex phenotypic responses. I have some concerns which I list below.

First, the authors should define what is meant by "aligning vernalization (or sometimes they use term vernalization response) to different climatic conditions". What exactly is the quantitative measure of the same? In the absence of a quantitative measure, how can one conclude which of the 3 distinct quantitative measures (initial levels, slopes) best explain the variation in "aligning vernalization"? Authors show in Figure 5 that initial levels of *FLC* and initial shut down show higher levels of variability and draw the conclusion higher levels of variability means higher association with vernalization… Is this assumption valid? Also, what is the coefficient of variation at final *FLC* levels? I am presuming these levels differ to between genotypes. Is this distinctly different from CoV of initial *FLC* levels?

From the data it appears the second aspect (*VIN3*-independent shut down) is primarily seen in Norwich climate and not much in other sites. How much of the initial *VIN3* induction explained by their temperature profiles?

I find the use of the "m-value" confusing. As far as I understood, the m-value appears to be highly variable independent of the allelic status of *FLC*. For example, Table 1 shows the m-value for Col to be 486 and Col FRI to be 36.5 (Norwich). Both genotypes have the same *FLC* allele. It is unclear to me how meaningful the conclusions of the m-value of *FLC* alleles are if the same allele has such different values? In this context, in the eighth paragraph of the Discussion, where the authors suggest that the variation in m-value derives from regulatory differences.

Reviewer #2:

The manuscript submitted by Hepworth et al. provides enormous data showing natural variation of vernalization response to the natural environments from Norwich to South and North Sweden, and suggests that starting level of *FLC* is critical for aligning vernalization to each climate. It is very difficult and time-consuming research to use natural environments with several accessions and mutants of Arabidopsis. In addition, to get some realistic idea about the evolutionary adaptive mechanism, we have to analyze the phenotypic response to natural environments. Thus, I think the work presented here is very useful information to not only plant scientists but also to the general readers. So, basically I recommend accepting this paper.

But I would like to suggest major revision as below. In that way, I think the manuscript will be easier to read. Otherwise, it will be too difficult to understand.

The major problem with this manuscript is the definition of the words.

For example, I could not understand the exact meaning of 'aligning vernalization response'. Could it be evolutionary fitness of vernalization? If it is, I think the authors should explain what kinds of phenotype are involved in the fitness (survival rate or seed number). Thus, I could not understand which data support the author's suggestion, 'starting *FLC* levels is critical for aligning vernalization to different climates'. I hope the manuscript revised to make it clear that how fitness and phenotypes of each accession are connected to the aligning vern.

Also, the meaning of *FLC*-post-vern value (m) should be clearly defined. I think this value is quite important to understand the whole manuscript. The value m does not look like *FLC* level right before transfer to warm temperature. As shown in Table 1, Col has much higher value than Col FRI (13 times higher). Thus, I feel there must be some equation to get the value and it must be provided.

Reviewer #3:

I have reviewed the manuscript entitled "Natural variation in autumn *FLC* levels, rather than epigenetic silencing, aligns vernalization to different climates" by Hepworth et al. This is an interesting manuscript dealing with an important topic in plant biology: the genetic mechanisms underlying flowering time, the most important fitness-related life-cycle trait in plants. Given the solid knowledge in the model plant *Arabidopsis thaliana* on the topic, the authors designed and carried out an array of experiments with various materials to address the role and interplay of *FLC* and *VIN3* genes in flowering time. The most remarkable and strongest aspect of this study has to do with the field experiments conducted over several years, which in my opinion is the right scenario to address all study questions in plant evolutionary genetics. Although I have enjoyed the manuscript, some conclusions seem not to be supported by the results or read too speculative. In addition, I consider that some methodological issues also require some clarification because they are affecting the interpretation of the results. Below I go through all my major concerns:

1) The Introduction could be developed further and make it more attractive to specialized and non-specialized readers. As it is, it reads too general and I do not understand why. It seems that the major motivation of this study is that "the importance of cis *FLC* polymorphism in delivering the different phases of *FLC* silencing in different climates (is) unknown". Given the extensive knowledge on the study system by the co-authors and the whole research community, this Introduction deserves a more extended development of the theoretical background, and most importantly, a working hypothesis to be put to the test. In other words, be ambitious (conceptually speaking) because this study deserves it. In addition, expression data from natural field settings is still rare in the literature, which should also be highlighted as a promising avenue to increase and improve our knowledge on the genetic mechanisms underlying flowering time.

2) It is said that one of the study years was unusually warm (Introduction and elsewhere). However, the reader does not know how unusual that year was in comparison with the historical weather records at the experimental facilities. Given that 10 of the hottest years ever recorded across the planet have been documented in the last 12-15 years, it is clear that we are experiencing an unstoppable warming (on top of many other evidences). The odds of having "unusual years" are high from now on. In any case, the authors could show deviances from historical weather records in a very simple manner to reinforce their point. I understand that weather data obtained from loggers are not from official weather data, but data from local meteorological stations for the study years could be compared to historical weather records (means and deviances). Actually, differences between data loggers and meteorological stations inform about micro-environmental variation at the experimental facilities, which could also be the case here.

3) The authors confound weather (or environment) by climate (Introduction and elsewhere). The authors are not comparing *FLC* expression between temperate and tropical climates, but across different weather or environmental conditions represented by the three experimental facilities (all of them within the same climate type). Please check it out throughout the text.

4) I am confused about the different sowing and moving (to field sites) dates for each experiment (subsection “Field experiments”). Ideally, the comparison among sites or years should be based on experiments established at the same time to avoid undesirable sources of variation, such as differences in environmental conditions (even in a period of weeks) and day length. I cannot assess how this problem affected the results because plants from different sites and years have been exposed to different environmental conditions during their development due to variation in sowing and moving dates (particularly for all the winter experiments). However, given one of the main results of this study (the uniform vernalisation response across haplotypes in different years and environments) the problems posed by this flaw were probably mitigated (I am speculating here). In any case and for the sake of clarity, the authors should explain the reasons of such sowing and moving schedule and the problems that such variability may cause on this sort of experiments.

5) My other major concern about this work is the treatment of fitness as a concept. The authors clearly state that survival and fecundity are the two major fitness components (subsection “Variation at *FLC* affects fitness in the field through branching and silique number”), the two traits are treated and analyzed separately. In my opinion, the authors have all the data to estimate fitness as the product between such two components, that is, survival (survival to seed set) by fecundity (number of fruits per plant). Expressions like "… silique production, and hence fitness, in surviving plants.…" should be avoided because they are not correct. In line with this, I consider branching as a secondary parameter that increases fruit fecundity. Somehow, the authors pay more attention to branching pattern than to survival throughout the manuscript (I do see the effects of *FLC* on it though). In any case, this work could gain clarity and consistency if the authors focus on how the interesting parameters found (*FLC* starting levels to name a relevant one) affect fitness. Then they will have the means to infer the evolutionary implications (in terms of local adaptation) of their results.

6) In relation to the previous point, I found that conclusions on the adaptive variation value of the results are speculative throughout the text, mainly because fitness has not been tackled appropriately. For example, the following sentence is quite obscure and hard to follow: "What we describe as the starting level was measured after some days in the field, so it is not equivalent to a non-vernalized control. Some *FLC* shutdown, most likely *VIN3*-independent, will have occurred at that time. Therefore, it is likely that the combination of these two determinants of *FLC* levels early in the field (starting levels, *VIN3*-independent shutdown) provides most of the potentially adaptive variation". I simply cannot see how the authors reach this conclusion, which deserves some work disentangling the effects of FLV and VIN-3 variation on fitness. There are further examples about this and overall the authors should tone-down the evolutionary implications of their results (see also "… our detailed analysis of the different phases of *FLC* silencing through winters in distinct climates, over multiple years, has given a clear picture of the mechanistic basis of adaptation in vernalization response”).

7) Finally, there are many style issues, particularly in the graphical material. For example, panels in Figure 1 should be lined up and have the same size and scale. Overall, I would increase symbol size to identify all haplotypes within panels and across figures or change the way to indicate that. Coefficients of variation are normally given as percentages. Letter size and type varies among figures. Chose the format (number of digits) to present data and be consistent (see the huge heterogeneity in Table 1). Crosses in Figure 6 are expected to be means (SE).

---

## [Author Response]

Essential revisions:While there was a common interest about the topic and the relevant findings, some major concerns on the description of methodologies and the interpretation of results as listed below need to be addressed to support the conclusion.1) The Introduction could be further improved with more attractive description of the background and a working hypothesis to be uniquely tested by expression data from field settings in this study.

We thank the editors and reviewers for this comment. We believe we have now significantly improved the Introduction to include a more full and attractive description as well as a working hypothesis.

2) The definition of "aligning vernalization response to different climatic conditions" is unclear. Clarification of the relevant phenotypes, the measures, and the parameters used for this analysis is necessary for understanding the conclusion drawn from this analysis.

We hope the extensive revisions we have made address this important issue.

3) The meaning of FLC-post-vern value (m) in Table 1 is confusing and should be clarified as understanding how this value is calculated and used in this study will considerably influence interpretation of the relevant results. In addition, how the constant related to non-FLC-mediated bolting delay is defined also should be explained.

We have re-written this paragraph to include the calculation in the main text as well as the table and supplementary figure legends and clarified that the constant is highly dependent on the specific experiment. A more detailed reply has been included in the response to individual reviewers.

4) The authors should explain the reasons of different sowing and moving (to field sites) schedules for each experiment, and discuss the potential problems pertaining to the results and conclusion that may result from such variability.

We now explain our motivation for sowing and moving for the Swedish sites (practicality and to ensure germination) in the Materials and methods, the reasons for the different sowing dates (local differences in climate) in the description of the field experiments, and the reasons for differences between years in the section ‘Natural variation in different phases of *FLC* silencing in the field’.

5) The authors should clarify with concrete evidence or correct some specific terms, such as "unusual years" and "climate", used in the manuscript.

Thank you, we have improved our definition of climate according to the Köppen-Geiger classification system, and shown comparison of temperature averages to support the observation of the unusually warm autumn in September 2016 in North Sweden (more detailed responses provided to the reviewers).

6) The conclusion and evolutionary implications of this study are not fully supported by the results based on the analysis of existing fitness components and their relationship with the major variable (FLC starting levels). The author should re-evaluate whether the traits (survival and fecundity) that were used as the major fitness components are appropriate for this study, and then correct some relevant descriptions and infer the conclusion and the relevant evolutionary implications accordingly.

We have changed the title of the last section to soften the claim. We think our explanation of our measures was principally at fault, and therefore we have changed the description of these results to clarify when we are discussing *FLC* levels, when we are discussing genotype fitness, and when we are discussing individual plant fitness. We have reduced our discussion of the branching factor, and changed the summing-up paragraph to say we have “shed light on” rather than “given a clear picture” to the mechanisms of adaptation.

7) As suggested by all reviewers, the quality of figures and tables as well as their citations in the text should be improved for style and clarity.

Thank you, we have made revisions in response to the individual suggestions from the reviewers, and would like to thank them for their help in substantially improving the manuscript.

Please take into consideration the specific comments from the reviewers below to revise the manuscript.Reviewer #1:[…] I have some concerns which I list below.First, the authors should define what is meant by "aligning vernalization (or sometimes they use term vernalization response) to different climatic conditions". What exactly is the quantitative measure of the same?

Prof. Balasubramanian raises a very good point – our definition is not clear. We have endeavoured to correct this, by stating that we took as ‘alignment’ the incidence of ‘appropriate’ or ‘springtime’ flowering as our measure, as in ‘aligning development with the seasons’. We have changed the title to remove the word ‘aligning’, and specify our hypothesis and measures more clearly, we hope, throughout the text.

In the absence of a quantitative measure, how can one conclude which of the 3 distinct quantitative measures (initial levels, slopes) best explain the variation in "aligning vernalization"? Authors show in Figure 5 that initial levels of FLC and initial shut down show higher levels of variability and draw the conclusion higher levels of variability means higher association with vernalization… Is this assumption valid?

We believe that the higher variability of the initial level and starting slopes is biologically most relevant because we see the greatest variation in resulting flowering time during the time at which the initial phase only is still ongoing (from transfer experiments in Figure 6A, C and E vs. B, D and F respectively). We have modified the manuscript to make this assumption clearer.

Also, what is the coefficient of variation at final FLC levels? I am presuming these levels differ to between genotypes. Is this distinctly different from CoV of initial FLC levels?

Prof. Balasubramanian raises an interesting and complex point. There are two obvious candidates for final *FLC* level – the level in spring, and the threshold that allows flowering in any given context. In the 2014-15 experiment, *FLC* levels in spring did not correlate with flowering time, likely because they were far below the point at which *FLC* had a detectable effect on flowering. We have now made this point in section “High autumnal FLC reduces precocious flowering in a warm autumn”.

Threshold is also difficult as figures Figure 6G and H shows that for almost all plants (except Lov1 and Var accessions) *FLC* levels are low enough to only delay, rather than totally prevent, flowering. Nevertheless, the *FLC* effects are quantitative over most of the range of expression during autumn. We therefore concentrated on autumn *FLC* dynamics, as at this time *FLC* was detectably delaying flowering in the transfer experiments, Figure 6, and in the field Figure 7. We have modified the text to make this clearer.

(N.B. in N Sweden 2016, we note that in the week preceding flowering, regardless of whether this was before or after winter, maximum daily temperatures were at least 15.5 and averaged 19.8°C, even though overall average daily temperatures were very cold (<10°C), strongly suggesting limitation by the ambient temperature pathway in parallel with that of the *FLC* effects, but that only part of the day need reach these temperatures to remove FLM/SVP repression).

From the data it appears the second aspect (VIN3-independent shut down) is primarily seen in Norwich climate and not much in other sites. How much of the initial VIN3 induction explained by their temperature profiles?

The *VIN3* induction is very sensitive to temperature. In previous work (Antoniou-Kourounioti et al., 2018) we showed that for Col *FRI* the temperature profile of these field studies could be used to predict *VIN3* expression using a mathematical model. Since the *VIN3* measurements follow similar patterns of response between the different genotypes at the same site (although with different maxima and minima), we would conclude the same complexity for the natural accessions and mutants presented in the current work. Comments were added to highlight the temperature sensitivity of *VIN3* in the manuscript (subsection “Natural variation in different phases of *FLC* silencing in the field”, third paragraph, Discussion, third paragraph).

I find the use of the "m-value" confusing. As far as I understood, the m-value appears to be highly variable independent of the allelic status of FLC. For example, Table 1 shows the m-value for Col to be 486 and Col FRI to be 36.5 (Norwich). Both genotypes have the same FLC allele. It is unclear to me how meaningful the conclusions of the m-value of FLC alleles are if the same allele has such different values? In this context, in the eighth paragraph of the Discussion, where the authors suggest that the variation in m-value derives from regulatory differences.

We have removed the ‘post-vern’ estimations for Col-0 as these are indeed confusing – the strange values are due to *FLC* being so low initially that the estimations are based on points that are too close together for accurate inference – reflected in the high p value (0.19) for the estimation. (In Col-0 flowering is mainly delayed by other pathways – other genotypes don’t reach similar levels of *FLC* until they have vernalised for a long time and pathways such as the miR156 pathway are no longer active). We have commented on this in the text.

We have rewritten the paragraph discussing the derivation of the *FLC* post-vern value to describe the linear regression in more detail and clarify that while in the accessions the effect is too much affected by other GxE effects to be reliable, within the NILs it is more consistent across environments.

Reviewer #2:[…] I would like to suggest major revision as below. In that way, I think the manuscript will be easier to read. Otherwise, it will be too difficult to understand.The major problem with this manuscript is the definition of the words.For example, I could not understand the exact meaning of 'aligning vernalization response'. Could it be evolutionary fitness of vernalization? If it is, I think the authors should explain what kinds of phenotype are involved in the fitness (survival rate or seed number). Thus, I could not understand which data support the author's suggestion, 'starting FLC levels is critical for aligning vernalization to different climates'. I hope the manuscript revised to make it clear that how fitness and phenotypes of each accession are connected to the aligning vern.

We thank the reviewer for the positive comments.

We have endeavoured to be clearer about our concept of ‘alignment’ throughout the text, by stating that this is the alignment of flowering with spring conditions, and that in North Sweden, where this alignment breaks down, both survival and reproduction are affected differently by the *FLC*haplotypes. We have therefore made several changes throughout the text.

Also, the meaning of FLC-post-vern value (m) should be clearly defined. I think this value is quite important to understand the whole manuscript. The value m does not look like FLC level right before transfer to warm temperature. As shown in Table 1, Col has much higher value than Col FRI (13 times higher). Thus, I feel there must be some equation to get the value and it must be provided.

We have tried to clarify this parameter by referring to the equation in the main text and reorganising this section. We have removed the Col-0 value (as mentioned in response to Reviewer 1) as, in the absence of FRI, *FLC* levels are too low to estimate the post-vern value accurately, and commented on this in the text.

Reviewer #3:I have reviewed the manuscript entitled "Natural variation in autumn FLC levels, rather than epigenetic silencing, aligns vernalization to different climates" by Hepworth et al. This is an interesting manuscript dealing with an important topic in plant biology: the genetic mechanisms underlying flowering time, the most important fitness-related life-cycle trait in plants. Given the solid knowledge in the model plant *Arabidopsis thaliana* on the topic, the authors designed and carried out an array of experiments with various materials to address the role and interplay of FLC and VIN3 genes in flowering time. The most remarkable and strongest aspect of this study has to do with the field experiments conducted over several years, which in my opinion is the right scenario to address all study questions in plant evolutionary genetics. Although I have enjoyed the manuscript, some conclusions seem not to be supported by the results or read too speculative. In addition, I consider that some methodological issues also require some clarification because they are affecting the interpretation of the results. Below I go through all my major concerns:1) The Introduction could be developed further and make it more attractive to specialized and non-specialized readers. As it is, it reads too general and I do not understand why. It seems that the major motivation of this study is that "the importance of cis FLC polymorphism in delivering the different phases of FLC silencing in different climates (is) unknown". Given the extensive knowledge on the study system by the co-authors and the whole research community, this Introduction deserves a more extended development of the theoretical background, and most importantly, a working hypothesis to be put to the test. In other words, be ambitious (conceptually speaking) because this study deserves it. In addition, expression data from natural field settings is still rare in the literature, which should also be highlighted as a promising avenue to increase and improve our knowledge on the genetic mechanisms underlying flowering time.

We thank the reviewer for the positive comments. We have developed the Introduction further and included specific hypotheses. We couldn’t possibly do much justice to the extensive work of the flowering time community within the word limit, but have tried to improve what we have. Regarding the major motivation – we only observed that the different phases were temperature-separable in these same experiments. We have rephrased the Introduction to make this clearer.

2) It is said that one of the study years was unusually warm (Introduction and elsewhere). However, the reader does not know how unusual that year was in comparison with the historical weather records at the experimental facilities. Given that 10 of the hottest years ever recorded across the planet have been documented in the last 12-15 years, it is clear that we are experiencing an unstoppable warming (on top of many other evidences). The odds of having "unusual years" are high from now on. In any case, the authors could show deviances from historical weather records in a very simple manner to reinforce their point. I understand that weather data obtained from loggers are not from official weather data, but data from local meteorological stations for the study years could be compared to historical weather records (means and deviances). Actually, differences between data loggers and meteorological stations inform about micro-environmental variation at the experimental facilities, which could also be the case here.

Many thanks for the suggestion, we have compared the September conditions for 2016 to weather-station data and found this year to be warm, although not the warmest in recent years – which is in itself interesting, as the reviewer notes. We have included the specific data in the “Natural variation in different phases of *FLC* silencing in the field” section.

3) The authors confound weather (or environment) by climate (Introduction and elsewhere). The authors are not comparing FLC expression between temperate and tropical climates, but across different weather or environmental conditions represented by the three experimental facilities (all of them within the same climate type). Please check it out throughout the text.

We thank Prof. Picó for pointing out that we have confused these concepts in some cases. We have corrected instances in which we were referring to the year-on-year differences in North Sweden, and further researched the classifications of climate to describe our field sites according to the Köppen-Geiger system of climate designation in the ‘Field experiments’ section of the Results.

4) I am confused about the different sowing and moving (to field sites) dates for each experiment (subsection “Field experiments”). Ideally, the comparison among sites or years should be based on experiments established at the same time to avoid undesirable sources of variation, such as differences in environmental conditions (even in a period of weeks) and day length. I cannot assess how this problem affected the results because plants from different sites and years have been exposed to different environmental conditions during their development due to variation in sowing and moving dates (particularly for all the winter experiments). However, given one of the main results of this study (the uniform vernalisation response across haplotypes in different years and environments) the problems posed by this flaw were probably mitigated (I am speculating here). In any case and for the sake of clarity, the authors should explain the reasons of such sowing and moving schedule and the problems that such variability may cause on this sort of experiments.

Indeed, this is a limitation on our work. The sowing and moving schedule in Sweden was used because of the difficulty of sowing directly into the field, and to ensure sufficient germination. A note has been made in the Materials and methods to explain this. For the different germination dates, reasons have been given in the main text, and a paragraph discussing this point in the Discussion.

5) My other major concern about this work is the treatment of fitness as a concept. The authors clearly state that survival and fecundity are the two major fitness components (subsection “Variation at FLC affects fitness in the field through branching and silique number”), the two traits are treated and analyzed separately. In my opinion, the authors have all the data to estimate fitness as the product between such two components, that is, survival (survival to seed set) by fecundity (number of fruits per plant). Expressions like "… silique production, and hence fitness, in surviving plants.…" should be avoided because they are not correct. In line with this, I consider branching as a secondary parameter that increases fruit fecundity. Somehow, the authors pay more attention to branching pattern than to survival throughout the manuscript (I do see the effects of FLC on it though). In any case, this work could gain clarity and consistency if the authors focus on how the interesting parameters found (FLC starting levels to name a relevant one) affect fitness. Then they will have the means to infer the evolutionary implications (in terms of local adaptation) of their results.

A previous version of this manuscript had investigated the effect of delayed flowering on mortality in our North Sweden transfers in more explicit detail, but the Editor and other reviewers suggested removing it, particularly as we do not know the cause of death of these plants (a point we have now added to the Discussion). Hence, we do not concentrate too much on *FLC* effects on survival. We have reduced the section on branching in the Discussion, and removed it from the Abstract. Within the final section, we have changed the language to make it clear when we are discussing individual plant fitness versus genotype fitness, and attempted to convey better how we have brought these elements together. We have changed the problematic sentence to:

“In the field, silique production is closely linked to branch production”

6) In relation to the previous point, I found that conclusions on the adaptive variation value of the results are speculative throughout the text, mainly because fitness has not been tackled appropriately. For example, the following sentence is quite obscure and hard to follow: "What we describe as the starting level was measured after some days in the field, so it is not equivalent to a non-vernalized control. Some FLC shutdown, most likely VIN3-independent, will have occurred at that time. Therefore, it is likely that the combination of these two determinants of FLC levels early in the field (starting levels, VIN3-independent shutdown) provides most of the potentially adaptive variation". I simply cannot see how the authors reach this conclusion, which deserves some work disentangling the effects of FLV and VIN-3 variation on fitness.

This was poorly phrased, we were not intending to imply an effect on fitness in this section, just highlighting that these metrics result in the different levels of *FLC* between the different accessions, and that if different levels of *FLC* have a fitness effect (which we investigate later) then these are the mechanisms that produce this variation. We have edited the text to clarify that the combination of these metrics “provides most of the variation in *FLC* levels”.

There are further examples about this and overall the authors should tone-down the evolutionary implications of their results (see also "… our detailed analysis of the different phases of FLC silencing through winters in distinct climates, over multiple years, has given a clear picture of the mechanistic basis of adaptation in vernalization response”.

We have modified our Abstract, conclusory statement within the Introduction, and Discussion accordingly.

7) Finally, there are many style issues, particularly in the graphical material. For example, panels in Figure 1 should be lined up and have the same size and scale.

We have corrected this.

Overall, I would increase symbol size to identify all haplotypes within panels and across figures or change the way to indicate that.

We have standardised the figure size and font size to ensure labels are clear. We have also changed the plots for flowering time to show the full distribution, which also makes the identification of each genotype easier.

Coefficients of variation are normally given as percentages.

We have corrected this.

Letter size and type varies among figures.

We have corrected this.

Chose the format (number of digits) to present data and be consistent (see the huge heterogeneity in Table 1).

We have done this for Table 1, thank you.

Crosses in Figure 6 are expected to be means (SE).

They are. We have noted that the error bars are s.e.m. In some cases errors are smaller than the crosses.